# Dietary fatty acids fine-tune Piezo1 mechanical response

Luis O. Romero [1], Andrew E. Massey[2], Alejandro D. Mata-Daboin[1], Francisco J. Sierra-Valdez[1,3,4], Subhash C. Chauhan [2], Julio F. Cordero-Morales [1] & Valeria Vásquez [1]

Mechanosensitive ion channels rely on membrane composition to transduce physical stimuli into electrical signals. The Piezo1 channel mediates mechanoelectrical transduction and regulates crucial physiological processes, including vascular architecture and remodeling, cell migration, and erythrocyte volume. The identity of the membrane components that modulate Piezo1 function remain largely unknown. Using lipid profiling analyses, we here identify dietary fatty acids that tune Piezo1 mechanical response. We find that margaric acid, a saturated fatty acid present in dairy products and fish, inhibits Piezo1 activation and poly-unsaturated fatty acids (PUFAs), present in fish oils, modulate channel inactivation. Force measurements reveal that margaric acid increases membrane bending stiffness, whereas PUFAs decrease it. We use fatty acid supplementation to abrogate the phenotype of gain-of-function Piezo1 mutations causing human dehydrated hereditary stomatocytosis. Beyond Piezo1, our findings demonstrate that cell-intrinsic lipid profile and changes in the fatty acid metabolism can dictate the cell's response to mechanical cues.

[1] Department of Physiology, College of Medicine, University of Tennessee Health Science Center, 71S. Manassas St., Memphis, TN 38163, USA. [2] Department of Pharmaceutical Sciences and Institute of Biomarker and Molecular Therapeutics (IBMT), College of Pharmacy, University of Tennessee Health Science Center, 881 Madison Ave., Memphis, TN 38163, USA. [3] Present address: Centro de Investigación Biomédica, Hospital Zambrano Hellion, TecSalud, Ave. Batallon de San Patricio 112, 66278 San Pedro Garza García, Nuevo León, Mexico. [4] Present address: Tecnólogico de Monterrey, Escuela de Ingeniería y Ciencias, Ave. Eugenio Garza Sada 2501 Sur, 64849 Monterrey, Nuevo León, Mexico. Correspondence and requests for materials should be addressed to V.V. (email: vvasquez@uthsc.edu)

M echanosensitive ion channels are membrane proteins that translate mechanical stimuli into electrical signals that lead to physiological responses. The Piezo1 mechanosensitive cation channel has been shown to mediate crucial physiological processes that include, but are not limited to, blood pressure regulation[1–3], vascular architecture and remodeling[4,5], erythrocyte volume regulation[6], chondrocyte mechanotransduction[7,8], lineage choice in neural stem cells[9], and association of neurons with astrocytes[10]. Piezo1 is important in mammals, as global knockouts are embryonic lethal in mice[4,5], and cell-specific knockouts result in animals with severe defects[11,12]. Cells continuously facing mechanical cues such as shear and osmotic stress, stretch, and changes in substrate stiffness rely on the fast response (milliseconds) of Piezo1 to regulate cellular processes that occur on larger timescales (seconds to days)[11,13]. Consequently, Piezo1 gain- and loss-of-function mutants have been associated with several hereditary human pathophysiologies[14,15]. For example, some mutations in Piezo1 are linked to inherited blood disorders (e.g., dehydrated hereditary stomatocytosis or xerocytosis[16–19]), in which the mutant channel displays slow inactivation[20], leading to an increase in cation permeability and subsequent dehydrated erythrocytes[12]. To better understand Piezo1 role in physiological and pathophysiological conditions, it is essential to determine the proteins and molecules that regulate its function.

While it is widely agreed that Piezo1 is activated by membrane tension[21–24], little is known about the role of membrane lipids[25,26]; particularly, whether fatty acids modulate Piezo1 function. Dietary fatty acids are among the membrane lipid components that dynamically regulate ion channel function[27,28]. The most common fatty acids in membrane lipids have 14–22 carbon atoms and carry one unsaturation (e.g., oleic acid), and in some cases as many as six double bonds[29]. These chemical features confer cells the capacity to control membrane thickness and fluidity in different physiological contexts. Polyunsaturated fatty acids (PUFAs), in particular, regulate plasma membrane remodeling by endowing the membrane with distinct elastic properties[27]. For instance, the function of TRP and TRP-like photomechanical receptors in *Drosophila melanogaster* is modulated by fatty acid-containing phospholipids, as increasing the content of saturated fatty acids in the diet leads to a decline in the electrical light response associated with these channels[30]. Likewise, plasma membranes containing the ω-3 PUFA eicosapentaenoic acid (EPA) and its eicosanoid derivative 17,18-EEQ enhance TRPV4 chemical and osmotic activities in human microvascular endothelial cells (HMVEC)[31]. Given that Piezo1 is activated by changes in membrane tension, we sought to determine whether changing the mechanical properties of the plasma membrane with different dietary fatty acids would have an effect on Piezo1 mechanical gating.

Here, we use lipidomics and electrophysiology to demonstrate that Piezo1-mediated mechanocurrents are modulated by dietary fatty acids in several cell lines including: the mouse neuro-2a (N2A) cells, HMVEC, and human embryonic kidney (HEK-293) cells. We find that when part of the plasma membrane, the saturated fatty acid margaric acid (MA; C17) inhibits Piezo1 currents, whereas C20 (arachidonic acid [AA] and EPA) and C22 (docosahexaenoic acid, DHA) PUFAs regulates inactivation. Differential scanning calorimetry (DSC) and atomic force microscopy (AFM) experiments show that membranes enriched in MA are more rigid and display higher bending stiffness than those containing PUFAs. N2A cells and HMVEC display distinct fatty acid profiles as well as Piezo1 different inactivation properties. We demonstrate here that culturing N2A cells with a fatty acid enriched in HMVEC changes Piezo1 to feature currents similar to the ones found in HMVEC. Finally, we determine that

fatty acids supplementation reverses the slow inactivation phenotype of Piezo1 gain-of-function mutations associated with human dehydrated hereditary stomatocytosis. Our findings demonstrate that saturated and PUFAs contained in the plasma membrane modulate the mechanical response of Piezo1.

## Results

**Margaric acid inhibits N2A endogenous Piezo1 currents.** Piezo1 channels were first identified by mechanically stimulating N2A cells using a piezo-electrically driven glass probe[32]. To determine how common dietary fatty acids modulate Piezo1's mechanical response in N2A cells, we characterized their membrane fatty acid profile using liquid chromatography-mass spectrometry (LC-MS). We found basal levels of the odd-chain saturated fatty acids MA (C17:0) and pentadecanoic acid (PDA; C15:0) (Supplementary Figure 1a, red). We hypothesized that increasing the amount of such fatty acids might alter membrane fluidity and in consequence modify Piezo1 activity. To this end, we performed whole-cell patch-clamp recordings (at −60 mV) after supplementing the N2A cell media with saturated fatty acids ranging between 1 and 300 μM for 18 h (overnight). These concentrations are within the range of circulating fatty acids present in blood plasma of healthy adults[33]. Piezo1 consistently displays robust and fast inactivating currents ($\tau \approx 33.8$ ms, at −60 mV) under mechanical displacement (Fig. 1a, control, and elsewhere[32]). Remarkably, we found that MA inhibits Piezo1 currents in a concentration-dependent manner (Fig. 1a, b), while retaining the time constants of inactivation of the non-treated cells (Supplementary Figure 1b). The magnitude of Piezo1 currents decreases with an $IC_{50} = 28.3 \pm 3.4$ μM (mean ± SEM; Fig. 1b). With high concentrations of MA (≥50 μM), the likelihood of activating Piezo1 decreases (Fig. 1a and Table 1 for 100 μM, 6 out of 13 cells did not display activity), even when poking the N2A cells to the limit of patch-clamp rupture.

We also used an alternative supplementation protocol consisting of adding lower doses of MA (insufficient to inhibit the channel) for several days. Supplementing N2A cells with 10 μM MA each day, for 5 days, significantly decreased Piezo1 currents (Fig. 1c). We used LC-MS to quantify the incorporation of MA in N2A membranes after fatty acid supplementation. The MA content increased 30- and 8-fold in cells treated with 100 μM MA overnight or 10 μM for 5 days, respectively, with respect to the control cells (Fig. 1d). On the other hand, supplementing N2A cells with PDA did not alter Piezo1 channel function (Supplementary Figure 1c–f) even though it was enriched 37-fold (Supplementary Figure 1g). As MA and PDA have similar structures but differ in their acyl-chain length (Fig. 1a and Supplementary Figure 1c), our results support the idea that saturated fatty acids with >15 carbons impair the mechanical response of Piezo1.

Next, we asked whether the inhibitory effect of MA is due to a decrease in Piezo1 channel function rather than a lack of channel expression or trafficking. Yoda1, known to sensitize Piezo1 to mechanical stimuli[34], recovered Piezo1 currents in MA (100 μM)-treated cells that could not be activated by displacement alone (Fig. 1a, e). This result supports the idea that MA shifts Piezo1 activation curve with no effect in channel expression or trafficking. Indeed, the displacement threshold for eliciting Piezo1 currents in MA (100 and 300 μM)-supplemented N2A cells was twofold when compared to that of the control (Fig. 1f). Although indentation generates variable stimulus-response relationships[11], we consistently required high displacement steps to elicit Piezo1 currents in the presence of MA. As expected, addition of Yoda1 prior mechanical stimulation lowered the displacement threshold, similar to the one obtained for the

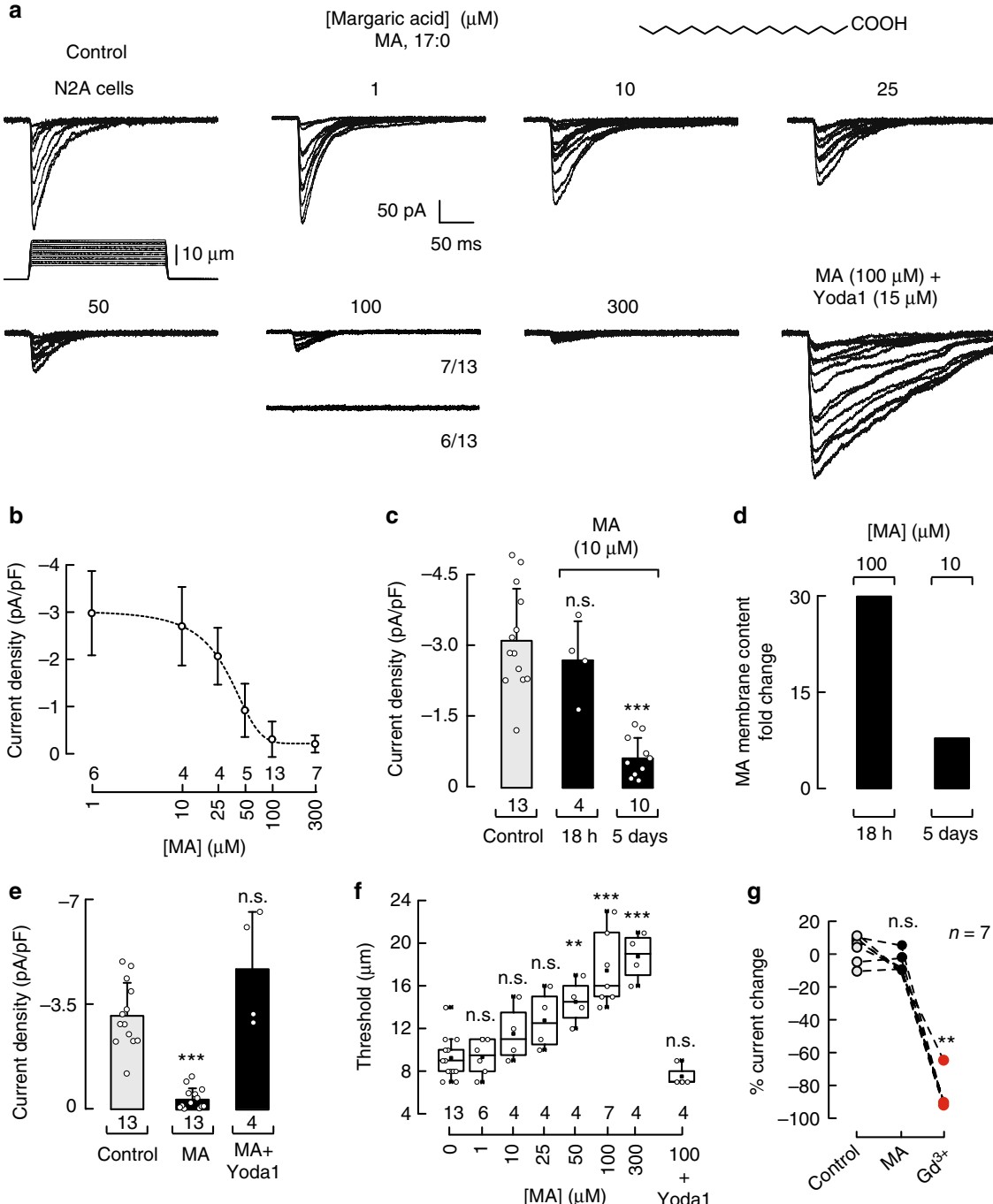

**Fig. 1** Margaric acid inhibits Piezo1 currents in N2A cells. **a** Representative whole-cell patch-clamp recordings (at −60 mV) of control and margaric acid (MA) (1, 10, 25, 50, 100, and 300 μM)-treated N2A cells elicited by mechanical stimulation. Bottom right panel displays representative Piezo1 macroscopic currents of MA-treated N2A cells incubated with Yoda1 prior to mechanical stimulation. The structure of MA is displayed on top. **b** Piezo1 current densities elicited by maximum displacement of MA-treated N2A cells. A Boltzmann function, Eq. (2), was fitted to the data (IC$_{50}$ = 28.3 ± 3.4 SEM). Circles are mean ± SD. $n$ is denoted above the $x$-axis. **c** Piezo1 current densities elicited by maximum displacement of control and MA (10 μM; 18 h and 5 days)-treated N2A cells. $n$ is denoted above the $x$-axis. Kruskal–Wallis and Dunn's multiple comparisons test. **d** MA membrane content fold change in N2A cells treated with MA 100 μM for 18 h or 10 μM each day for 5 days, as determined by LC-MS. **e** Piezo1 current densities elicited by maximum displacement of control, MA (100 μM)-treated N2A cells, and MA (100 μM)-treated N2A cells incubated with 15 μM Yoda1 prior to mechanical stimulation. Bars are mean ± SD. $n$ is denoted above the $x$-axis. Kruskal–Wallis and Dunn's multiple comparisons test. **f** Boxplots show the mean, median, and the 75th to 25th percentiles of the displacement thresholds required to elicit Piezo1 currents of control and MA-treated N2A cells. n is denoted above the $x$-axis. One-way ANOVA and Bonferroni test. **g** Current changes elicited by maximum displacement of N2A cells perfused for 120 s with bath solution (control), MA (100 μM), and Gd$^{3+}$ (30 μM) consecutively. Gd$^{3+}$ was used as control for the perfusion. Data samples are paired. $n$ is denoted above the plot. Friedman test. Asterisks indicate values significantly different from control (***$p < 0.001$ and **$p < 0.01$) and n.s. indicates not significantly different from the control

control (Fig. 1f). Interestingly, perfusion of 100 μM MA in solution to N2A cells while poking under the whole-cell configuration did not inhibit Piezo1 currents (Fig. 1g and Supplementary Figure 1h), suggesting that the effect of MA could be due to membrane remodeling rather than a direct effect.

**Table 1 Proportion of N2A cells that display Piezo1 currents after MA supplementation**

| [Margaric acid] (μM) | % of N2A cells with Piezo1 currents |
| --- | --- |
| 0 | 100 |
| 1 | 100 |
| 10 | 100 |
| 25 | 100 |
| 50 | 80 |
| 100 | 54 |
| 300 | 57 |
| 100 + Yoda1 | 100 |

Taken together, our results demonstrate that MA treatment inhibits Piezo1 currents by increasing the mechanical threshold required for activation.

**PUFAs modulate Piezo1 inactivation.** Long ω-6 and ω-3 PUFAs (≥C20)—enriched in meat and fish, respectively—are known to increase membrane structural disorder, decrease stiffness, and alter ion channel function[27,31,35–38]. To determine the role of long PUFAs in Piezo1 mechanical gating, we supplemented the N2A cell media with ω-6 AA (C20:4), ω-3 EPA (C20:5), and ω-3 DHA (C22:6). We found that C20 PUFAs decreased Piezo1 time constants of inactivation to ≈20 ms (100 μM of AA and EPA, Fig. 2a, b) compared to control cells (≈34 ms). Moreover, we observed a further decrease in the time constants of inactivation (≈13 ms) when supplementing N2A cells with higher EPA concentrations (200 and 300 μM, Supplementary Figure 2a, b). Interestingly, supplementing N2A cells with low concentrations (10 μM) of EPA, each day for 5 days, increases the EPA content in the plasma membrane and significantly speed up Piezo1

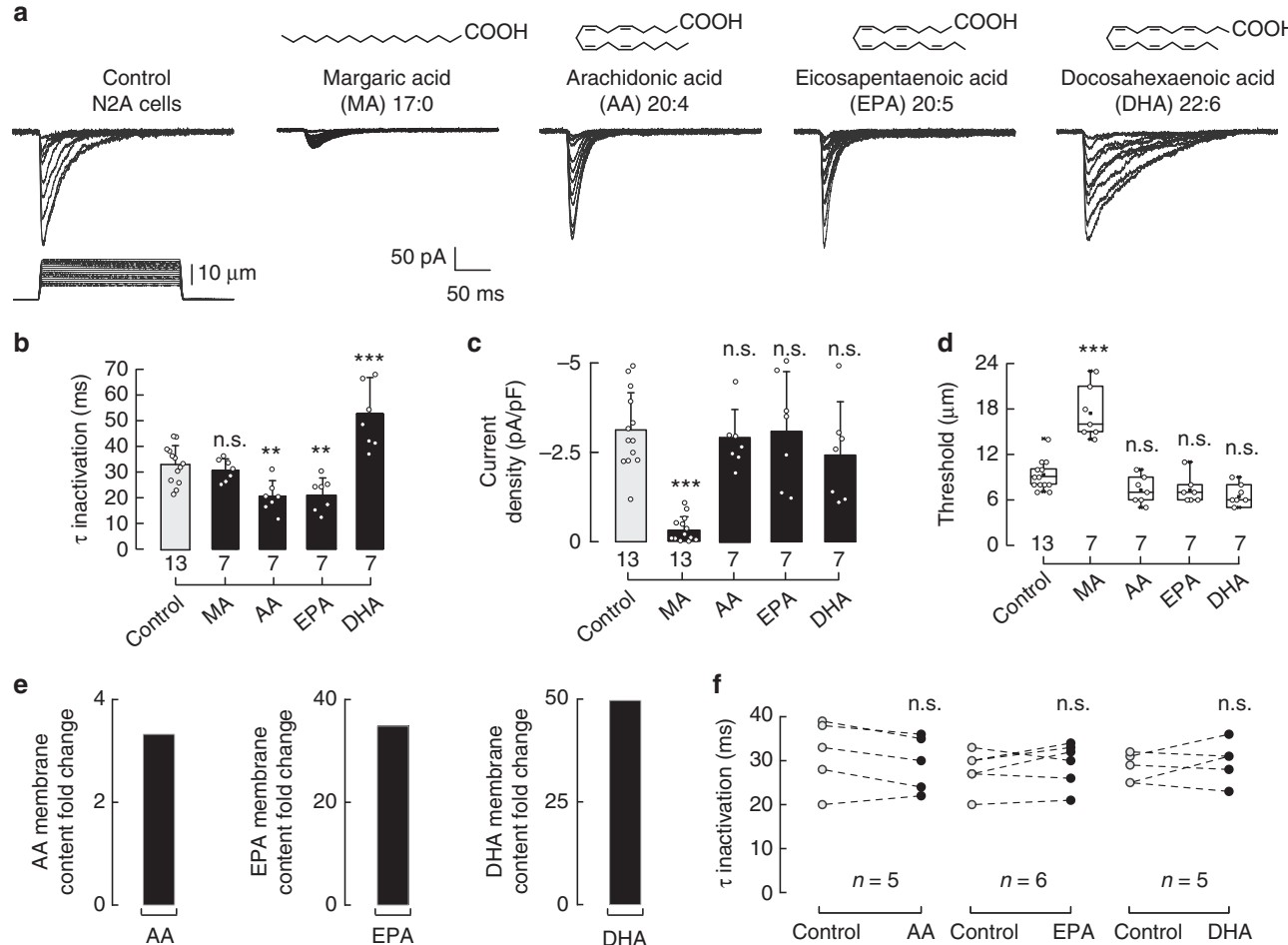

**Fig. 2** Dietary fatty acids alter Piezo1 channel gating. **a** Representative whole-cell patch-clamp recordings (at −60 mV) of control, MA, AA, EPA, and DHA (100 μM)-treated N2A cells elicited by mechanical stimulation. The structures of MA, AA, EPA, and DHA are displayed above. **b** Piezo1 time constants of inactivation elicited by maximum displacement of control, MA, AA, EPA, and DHA (100 μM)-treated N2A cells. Bars are mean ± SD. One-way ANOVA and Bonferroni test. **c** Current densities elicited by maximum displacement of control, MA, AA, EPA, and DHA (100 μM)-treated N2A cells. Bars are mean ± SD. Kruskal–Wallis and Dunn's multiple comparisons test. **d** Boxplots show the mean, median, and the 75th to 25th percentiles of the displacement threshold required to elicit Piezo1 currents of control, MA, AA, EPA, and DHA (100 μM)-treated N2A cells. One-way ANOVA and Bonferroni test. **e** AA, EPA, and DHA membrane content fold change in N2A cells treated with AA, EPA, and DHA 100 μM for 18 h, as determined by liquid chromatography-mass spectrometry (LC-MS). **f** Piezo1 time constants of inactivation elicited by maximum displacement of N2A cells perfused for 120 s with bath solution (control) and a PUFA (AA, EPA, and DHA; 100 μM). Data samples are paired. Paired t-test. Asterisks indicate values significantly different from control (***p < 0.001 and **p < 0.01) and n.s. indicates not significantly different from the control. n is denoted above the x-axes

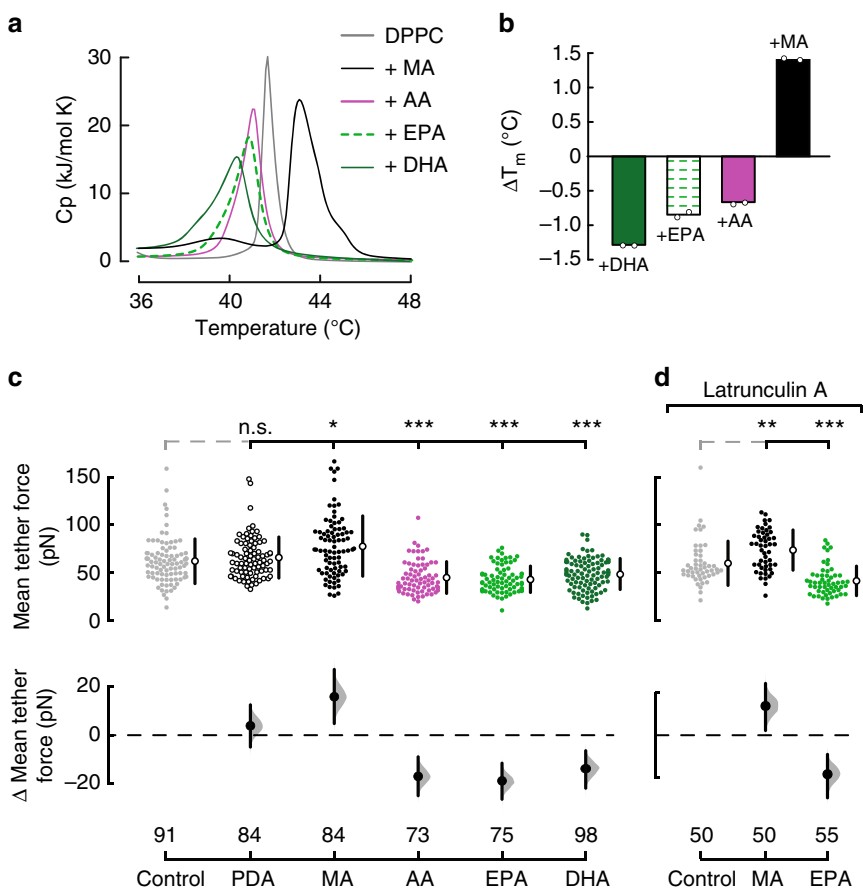

**Fig. 3** Dietary fatty acids alter membrane fluidity in N2A cells. **a** Representative thermotropic characterization of the DPPC/fatty acid systems using DSC: control ($T_m = 41.68\,°C$), MA (43.08 °C), AA (41.04 °C), EPA (40.85 °C), and DHA (40.40 °C). **b** Effects of DPPC/fatty acids on melting temperatures ($\Delta T_m$) with respect to DPPC membranes. Experiments were performed from two independent preparations. **c** Estimation plot with all tether force data points presented as a swarmplot for control, pentadecanoic acid (PDA), MA, AA, EPA, and DHA (100 μM)-treated N2A cells. **d** Estimation plot with all tether force data points presented as a swarmplot for latrunculin A (1 μM; 1 h)-treated N2A control cells and supplemented with MA (100 μM) or EPA (100 μM). Bootstraps with 99.9% confidence interval and a comparison with the mean of the control group are displayed to the right and bottom of the raw data, respectively. Kruskal–Wallis and Dunn's multiple comparisons test. Asterisks indicate values significantly different from control (***$p < 0.001$, **$p < 0.01$, and *$p < 0.05$) and n.s. indicates not significantly different from the control. $n$ is denoted above the x-axes

inactivation (≈13 ms; Supplementary Figure 2a–c). On the other hand, C22 DHA elicited a robust increase in the time constants of inactivation up to ≈53 ms (100 μM, Fig. 2a, b). We did not find differences in current magnitudes or displacement thresholds with these long PUFAs when compared to those of the control (Fig. 2c, d). It is remarkable that the ω-3 fatty acids EPA and DHA elicited distinct effects on inactivation even though their structures are similar (Fig. 2a, top). We also determined that PUFAs (AA, EPA, and DHA) are efficiently incorporated in N2A membranes (Fig. 2e). Our results suggest that saturated fatty acids (like MA) control the extent of Piezo1 activation, and once Piezo1 is open long PUFAs determine the time course of its inactivation. Similar to MA, when directly perfused while mechanically stimulating N2A cells, long PUFAs did not have an effect on Piezo1 inactivation or current magnitudes (Fig. 2f and Supplementary Figure 2d). Hence, we hypothesized that fatty acids modulate Piezo1's activity by changing the physical properties of the plasma membrane (e.g., structural disorder and stiffness).

**Fatty acids alter Piezo1 via the plasma membrane.** Using DSC, we measured changes in heat-capacity profiles ($C_p$) as a readout for membrane structural disorder of synthetic membranes (1,2-dipalmitoyl-sn-glycero-3-phosphocholine, DPPC) with and without fatty acids. We found that MA evokes a robust shift in the melting temperature ($T_m$) toward higher values indicative of an increase in membrane structural order ($\Delta T_m = 1.39$; Fig. 3a, b, black). On the other hand, long PUFAs increase membrane structural disorder and decrease cooperativity among lipids, as observed by the leftward shift and broadening of the main melting peaks corresponding to DHA, EPA, and AA (Fig. 3a, b, dark and light greens, and magenta, respectively). The DSC results in synthetic lipids suggest that fatty acids endow membranes with distinct structural properties that might determine the magnitude and extent of Piezo1's mechanical response.

Measurements of tether force using AFM allow determination of the effective plasma membrane fluidity and bending stiffness in live cells[31,37]. We have previously determined that plasma membranes enriched in EPA feature lower bending stiffness than control cells[31,37]. To directly determine the extent by which PDA, MA, AA, EPA, and DHA alter the mechanical properties of N2A plasma membranes, we measured the mean tether forces of control and fatty acid-treated cells. The mean tether force for control N2A cells (61.9 ± 23.4 pN, mean ± SD; Fig. 3c) is similar to the ones reported for other cultured cells and neurons[31,37,39]. We found that MA treatment significantly increased the mean tether force (77.8 ± 31.5 pN, mean ± SD) when compared to untreated cells (Fig. 3c and Supplementary Figure 3a), indicating

that plasma membranes enriched in MA display high bending stiffness. Importantly, treatment with PDA did not alter the elastic properties of N2A plasma membranes ($65.8 \pm 21.5$; mean $\pm$ SD; Fig. 3c); this property correlates with its inability to modify Piezo1 mechanical response (Supplementary Figure 1c–f). On the other hand, C20 and C22 PUFAs had the opposite effect (low bending stiffness) on plasma membranes as they significantly decreased their mean tether force ($44.9 \pm 16.6$ and $43 \pm 13.8$, $48.5 \pm 16.2$, mean $\pm$ SD, AA, EPA and DHA, respectively; Fig. 3c).

To evaluate the contribution of the cytoskeleton to the mean tether force of membranes enriched with specific fatty acids, we disrupted actin polymerization using latrunculin A. Cells treated with latrunculin A display round shape with less filopodia like-protrusions (Supplementary Figure 3b). We found that this treatment did not have an effect on the mean tether forces of N2A plasma membranes ($61.9 \pm 23.4$ control vs. $59.7 \pm 23.1$ latrunculin A, mean $\pm$ SD; Fig. 3c, d, light gray circles). Moreover, disrupting the actin filaments did not change the effect that fatty acids enrichment had on the plasma membrane mechanics, as MA increased the mean tether force ($73.6 \pm 21$, mean $\pm$ SD) and EPA decreased it ($41.3 \pm 15.4$, mean $\pm$ SD; Fig. 3d). As expected, in latrunculin A-treated cells, MA inhibits Piezo1 currents and increases the displacement threshold, and EPA enhances its inactivation (Supplementary Figure 3c–f). Interestingly, disrupting actin polymerization decreases the magnitude of Piezo1 current densities ($-3.11 \pm 1.1$ control vs. $-1.49 \pm 0.5$ latrunculin A-treated N2A cells, mean $\pm$ SD; Fig. 1c and Supplementary Figure 3d, gray bars). This result suggests that less channels are being recruited with the mechanical probe in cells treated with latrunculin A, likely due to the disrupted cytoskeleton and an inefficient transmission of the mechanical stimuli. Similar results were reported by the Goodman laboratory when measuring mechanoreceptor currents and mean tether forces from neurons lacking cytoskeletal proteins[40,41]. Altogether, our results support that MA decreases Piezo1 activation by increasing membrane structural order and bending stiffness, whereas PUFAs modify Piezo1 inactivation by decreasing structural order and bending stiffness.

**Piezo1 mediates indentation-activated currents in HMVEC**. Piezo1 is also expressed in endothelial cells, where it has been shown to gate in response to shear stress[4,5], enable vascular development[4,5], mediate flow-induced ATP release[1], and sense physical activity[2]. Whether Piezo1 mediates indentation-activated currents in HMVEC has not been previously established. HMVEC have a fatty acid distribution profile different from that of N2A cells, highly enriched in saturated fatty acids and ω-6 PUFAs (Fig. 4a, gray and magenta), as determined by LC-MS. In view of the results described above, we anticipated that Piezo1 in HMVEC would feature channel properties distinct from those measured in N2A cells. To test this idea, we characterized the mechanically activated currents in HMVEC by indenting the cells with 1-μm-step pulses. We found that HMVEC indentation-activated currents (Fig. 4b) displayed most of the properties described for Piezo1 in other cell types: voltage-dependent inactivation[42], non-selective cation currents[32] (reversal potential at $+7.5$ mV, Fig. 4c), complete blockage by $GdCl_3$[32], and partial inhibition by L-GsMTX-4[20] (Supplementary Figure 4a). Moreover, specific blockers of other cation channels (e.g., TRPC3, TRPC6, TRPV4, and TRPA1) did not decrease the magnitude of the indentation-mediated currents (Supplementary Figure 4b). It is noteworthy that the large steady-state currents ($\approx 50\%$ of the peak, red arrow) observed in most of our whole-cell recordings seem to be a feature for HMVEC mechano-activated currents

(Fig. 4b). Furthermore, we determined that Piezo1 is functional in HMVEC when measuring a significant increase in the fluorescence signal after addition of Yoda1 to cells loaded with a $Ca^{2+}$-sensitive dye (Fluo-4 AM; Fig. 4d and Supplementary Figure 5a). Finally, knocking down the expression of Piezo1 with silencing RNA (siRNA) abolished, almost completely, the indentation-mediated currents of HMVEC (Fig. 4e and Supplementary Fig. 5b–f). Overall, our results demonstrate that Piezo1 mediates indentation-activated currents in HMVEC and displays distinct mechano currents likely due to differences in the fatty acids profile.

**Fatty acids alter HMVEC Piezo1 channel gating**. We asked whether Piezo1 gating modulation by fatty acids in N2A cells was only a feature of this mouse cell line or could occur in other cell types. To this end, we measured the effects of MA and long PUFAs on Piezo1 activity under an inherently different fatty acid regime like the one found in HMVEC (Fig. 4a). As observed in N2A cells, MA supplementation in HMVEC inhibited Piezo1 currents (Fig. 5a, b) by increasing the displacement threshold (Fig. 5c). Moreover, MA treatment did not alter the ratio of peak to steady-state currents (i.e., inactivation) of Piezo1 (Fig. 5d). In contrast to MA, C20 and C22 PUFAs have differential effects on Piezo1 inactivation in HMVEC. EPA (100 μM; 18 h) slightly diminishes (albeit not significant) the ratio of peak to steady-state currents (Fig. 5a, d); this result was unexpected since EPA strongly enhances the inactivation in N2A cells. One explanation for this result might be that HMVEC membranes require more EPA than N2A cells to modulate Piezo gating. We tested this idea by supplementing HMVEC with low EPA concentrations for several days (50 μM; 3 days). Indeed, using this supplementation protocol, we found an increase in EPA content (18 h vs. 3 days, Fig. 5e) in HMVEC as well as a significant increase in inactivation (Fig. 5f). Notably, PUFAs did not change current densities or the displacement thresholds (Fig. 5b, c and Supplementary Figure 6a, b). As observed in N2A cells, DHA decrease inactivation in HMVEC (Fig. 5a, d), as the currents at the end of the indentation pulse were $\approx 85\%$ of the activation peak, while maintaining current densities and displacement thresholds similar to those of the control cells (Fig. 5b, c). These results demonstrate that Piezo1 modulation by fatty acids is not unique to N2A cells and is independent of the channel ortholog (mouse vs. human). LC-MS analysis demonstrates that in the culture conditions used for electrophysiological recordings, HMVEC are able to incorporate MA, AA, EPA, and DHA in the membranes, albeit to different extents (Fig. 5e, g). Altogether our results, from N2A and HMVEC, show that supplementation of saturated and unsaturated fatty acids could be useful for tuning cells' responses to mechanical stimuli.

**Global fatty acid distribution modulates Piezo1 gating**. When analyzing the fatty acid content of N2A cells and HMVEC, we noticed that linoleic acid (C18:2) was higher in HMVEC (Fig. 6a). Linoleic acid – an essential ω-6 fatty acid – is a metabolic precursor for AA; due to its limited conversion by the delta-6 desaturase enzyme[43], linoleic acid can accumulate when its consumption in the diet is increased. Indeed, we observed accumulation of linoleic acid in supplemented N2A cells, while keeping the content of downstream AA lower than the precursor (Supplementary Figure 7a, right panel). As Piezo1 inactivation in N2A cells is more prominent than in HMVEC, we wondered whether increasing the content of linoleic acid and its downstream metabolic products would modify Piezo1's mechanical response to resemble that of HMVEC. Remarkably, N2A cells supplemented with linoleic acid displayed a global fatty acid

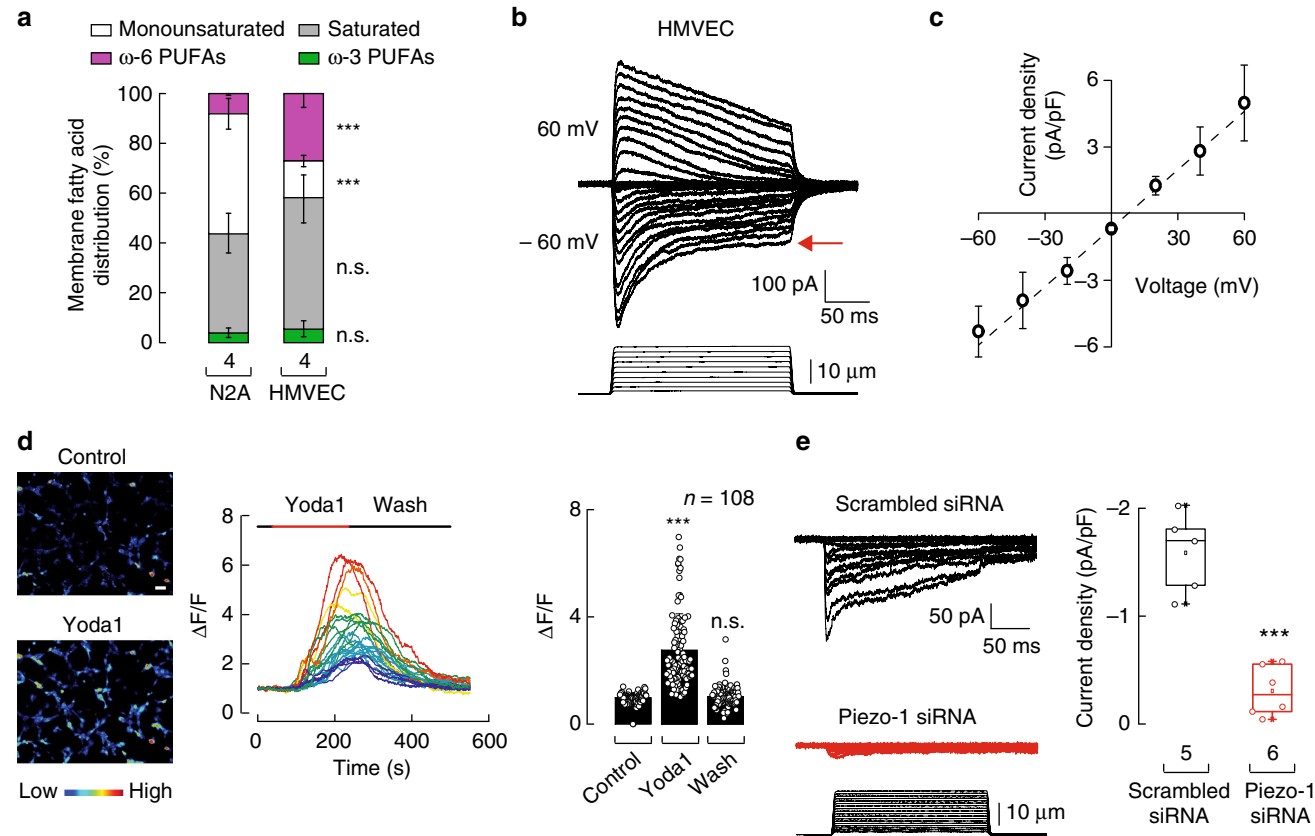

**Fig. 4** Piezo1 channel mediates mechanically evoked currents in HMVEC. **a** Stacked bar chart illustrating the membrane fatty acid distribution in N2A cells and HMVEC, as determined by LC-MS. Bars are mean ± SD. $n$ is denoted above the $x$-axis. Mann–Whitney test. **b** Representative whole-cell patch-clamp recordings (at ±60 mV) of HMVEC (top) elicited by mechanical stimulation (bottom). Red arrow highlights the steady-state currents. **c** Current–voltage relationship of HMVEC mechano-dependent currents as determined by whole-cell patch-clamp experiments. Circles are mean ± SD. $n = 5$. **d** Left: representative micrographs of HMVEC challenged with control buffer and 2 μM Yoda1 and analyzed for their responses using $Ca^{2+}$ imaging (Fluo-4 AM); color bar indicates relative change in fluorescence intensity. White bar represents 50 μm. Middle: representative traces corresponding to intensity changes ($\Delta F/F$) of individual cells shown in left panel. Right: mean fluorescence intensity values ($\Delta F/F$) of HMVEC perfused with control solution ($t = 30$ s), Yoda1 ($t = 250$ s), and washed with control solution ($t = 500$ s). Bars are mean ± SD. $n$ is denoted above the plot. Friedman test. **e** Left: representative HMVEC mechano-dependent current densities transfected with scrambled or Piezo1 siRNA elicited by mechanical stimulation at −60 mV under the whole-cell patch-clamp configuration. Right: boxplots show the mean, median, and the 75th to 25th percentiles of mechano-dependent currents densities obtained by whole-cell patch-clamp recordings of scrambled or Piezo1 siRNAs transfected HMVEC. $n$ is denoted above the $x$-axis. Unpaired $t$-test. Asterisks indicate values significantly different from control (***$p < 0.001$) and n.s. indicates not significantly different from the control

distribution profile similar to that of HMVEC (i.e., saturated fatty acids and ω-6 PUFAS become more abundant; Fig. 6b). Under these conditions, we found a decrease in Piezo1 inactivation (Fig. 6c, d) and steady-state currents reminiscent of the ones measured in HMVEC (Fig. 6c, black arrow, and 6d). As AA enhances Piezo1 inactivation in N2A cells (Fig. 2a, b), we propose that the effect on inactivation in linoleic acid-treated cells might be due to an overall increase in linoleic acid itself and/or downstream ω-6 PUFAs rather than AA (Fig. 6b). Taken together, our data support the idea that distinct Piezo1 gating properties, like the ones reported for N2A cells and HMVEC, are determined, in part, by the cellular fatty acid permutation.

**EPA abrogates the phenotype of mutations causing xerocytosis.** Mutations in human Piezo1 are associated with dehydrated hereditary stomatocytosis, a hemolytic anemia characterized by an increased cation permeability and a decreased osmotic fragility of erythrocytes[16,17,44]. For example, the alleles R1943Q[18], M2225R[16–18], R2302H[18], R2456H[16,17,45], and R2488Q[18,19] exhibit decreased inactivation when compared to wild type. In the

mouse Piezo1 structure[46], the equivalent positions to these residues are mapped in the blade (R1943), extracellular cap (M2225 and R2302), pore (R2456), and C-terminal (R2488) domains, suggesting that several regions might contribute to the inactivation process (Fig. 7a). Molecules that enhance channel inactivation or decrease activation could be useful to diminish cation permeability and erythrocyte fragility in this pathophysiological context. Given that EPA significantly decreased Piezo1's time constant of inactivation in N2A cells and HMVEC (Figs. 2b and 5e), we sought to determine whether EPA supplementation can recover Piezo1 wild-type inactivation in the frame of these slow inactivating mutants. As reported by other groups[16–19,45], we found that these mutants display decreased inactivation while keeping similar displacement thresholds than the wild type (Fig. 7b–d); with the exception of mutants R2302H and R2488Q that required less displacement to activate (Fig. 7d). Remarkably, EPA treatment speeds up the time constant of inactivation of Piezo1 mutants to levels that resemble that of the wild-type channel (Fig. 7b, right panel, and Fig. 7c); as expected, EPA does not alter the activation threshold of the mutants (Fig. 7d). We transfected Piezo1 and mutants M2225R and R2456H (that

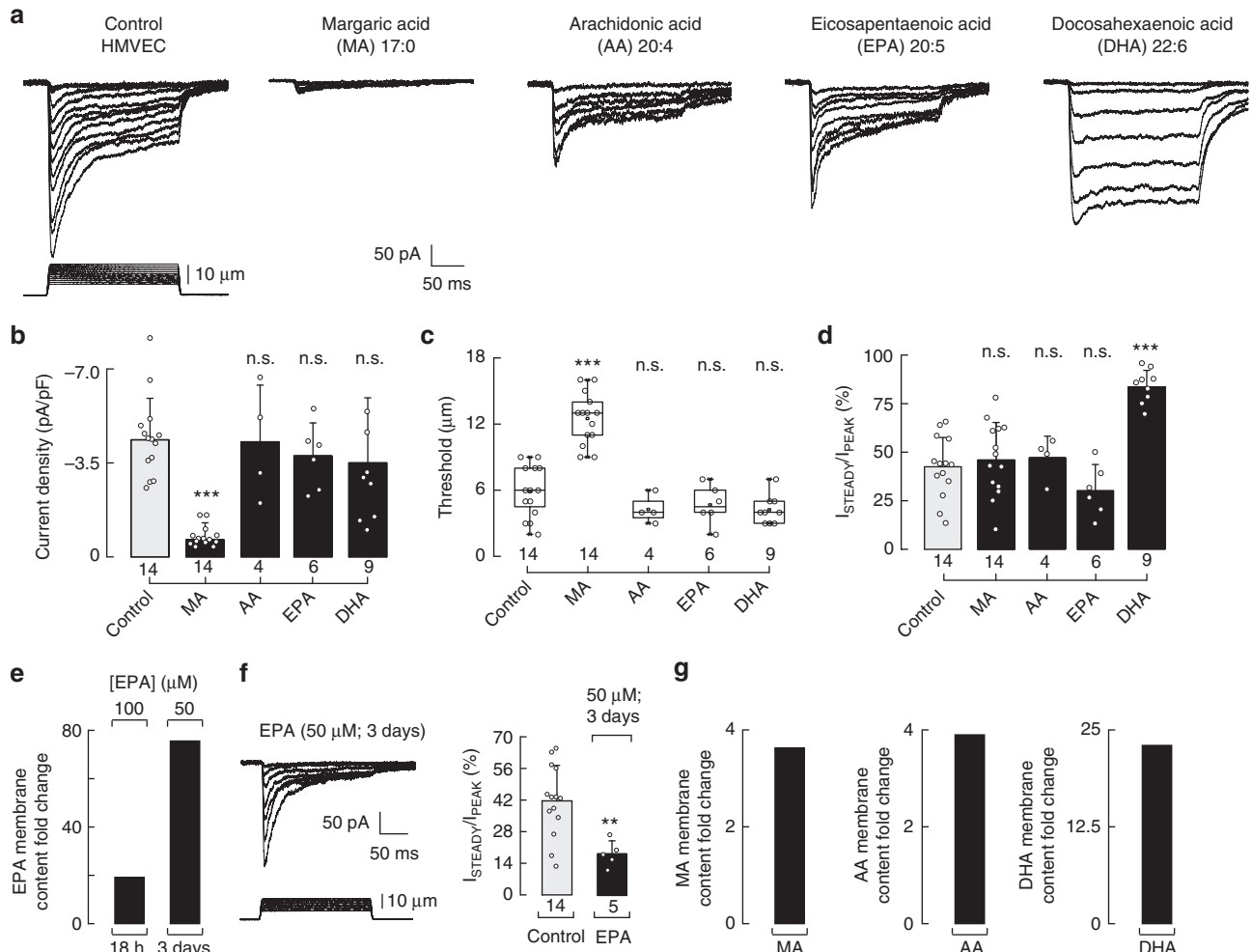

**Fig. 5** Dietary fatty acids alter Piezo1 channel gating in HMVEC. **a** Representative whole-cell patch-clamp recordings (at −60 mV) of control, MA, AA, EPA, and DHA (100 μM)-treated HMVEC elicited by mechanical stimulation. **b** Piezo1 current densities elicited by maximum displacement of control, MA, AA, EPA, and DHA (100 μM)-treated HMVEC. Bars are mean ± SD. Kruskal–Wallis and Dunn's multiple comparisons test. **c** Boxplots show the mean, median, and the 75th to 25th percentiles of the displacement threshold required to elicit Piezo1 currents of control and MA, AA, EPA, and DHA (100 μM)-treated HMVEC. One-way ANOVA and Bonferroni test. **d** Ratio of the currents at the end of the displacement pulse to the peak current ($I_{STEADY}/I_{PEAK}$) from macroscopic traces of control, MA, AA, EPA, and DHA (100 μM)-treated HMVEC. Bars are mean ± SD. One-way ANOVA and Bonferroni test. **e** EPA membrane content fold change of EPA-treated HMVEC supplemented with 100 μM for 18 h and 50 μM each day for 3 days, as determined by LC-MS. **f** Representative whole-cell patch-clamp recordings (at −60 mV) of EPA (50 μM each day for 3 days)-treated HMVEC elicited by mechanical stimulation; and ratio of the currents at the end of the displacement pulse to the peak current ($I_{STEADY}/I_{PEAK}$) from macroscopic traces of control and EPA (50 μM each day for three days)-treated HMVEC. Bars are mean ± SD. Unpaired $t$-test. **g** MA, AA, and DHA membrane content fold change in MA, AA, and DHA (100 μM)-treated HMVEC as determined by LC-MS. Asterisks indicate values significantly different from control (***$p < 0.001$ and **$p < 0.01$) and n.s. indicates not significantly different from the control. $n$ is denoted above the x-axes

display the slowest inactivation time constants) in HEK-293 cells, to test whether EPA supplementation had the same effect in another cell type. As determined in N2A cells, EPA treatment decreases the time constant of inactivation in the wild-type and mutant channels expressed in HEK-293 cells (Supplementary Figure 8b, c); EPA does not alter the activation threshold or current densities of the mutants (Supplementary Figure 8d–e). Altogether, these results suggest that enriching the plasma membrane with EPA overrides the effect of these gain-of-function mutations.

**Fatty acids have synergistic effects on Piezo1 function**. Our previous results demonstrate that MA modulates Piezo1 channel activation, whereas PUFAs alter channel inactivation. We next asked whether a combination of saturated and unsaturated fatty

acids would have a synergistic effect on Piezo1 activity. To test this idea in endogenous Piezo1, we supplemented N2A cells with EPA and MA and found a combined effect on Piezo1 gating as the currents inactivated faster, were smaller, and needed more displacement for activation than the control cells (Fig. 8a–d). As dehydrated hereditary stomatocytosis mutants increased $Ca^{2+}$ influx in erythrocytes, we tested the ability of the fatty acid combination to modulate the response of the gain-of-function R2456H mutant. Notably, EPA and MA supplementation reduced the inactivation time constant and current densities while increasing the displacement threshold of the R2456H mutant (Fig. 8e–h). Blood plasma is a high-protein buffer as it contains ~70 g L⁻¹ of proteins; some of which bind to fatty acids. We supplemented N2A cells culture media with 70 g L⁻¹ of BSA (the most abundant protein in human blood plasma) and found that in these conditions EPA supplementation was also efficient in

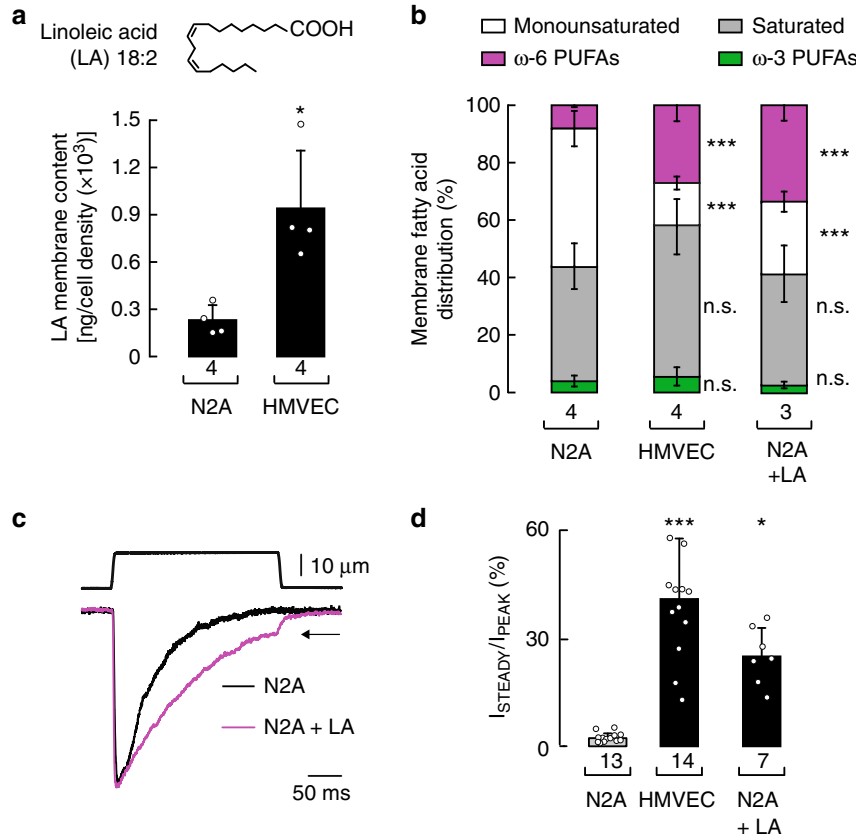

**Fig. 6** Changing the cellular fatty acid distribution profile modifies Piezo1 inactivation. **a** Linoleic acid (LA) membrane content in N2A cells and HMVEC, as determined by LC-MS. Bars are mean ± SD. Mann–Whitney test. **b** Stacked bar chart illustrating the membrane fatty acid distribution in N2A cells, HMVEC, and LA (100 μM)-treated N2A cells, as determined by LC-MS. Bars are mean ± SD. One-way ANOVA and Bonferroni test. **c** Representative whole-cell patch-clamp recordings (at −60 mV) of N2A cells and linoleic acid (LA, 100 μM)-treated N2A cells elicited by mechanical stimulation. Traces were normalized for comparison. **d** Ratio of the currents at the end of the displacement pulse to the peak current ($I_{STEADY}/I_{PEAK}$) from macroscopic traces of N2A cells, LA (100 μM)-treated cells, and HMVEC. Bars are mean ± SD. Kruskal–Wallis and Dunn's multiple comparisons test. Asterisks indicate values significantly different from control (***$p < 0.001$ and *$p < 0.05$) and n.s. indicates not significantly different from the control. $n$ is denoted above the $x$-axis

enhancing Piezo1 inactivation (Supplementary Fig. 8f, g). As EPA and MA are biosynthesized and commonly found in fish and dairy products[47,48], respectively, individual or combined fatty acids could be used as a dietary strategy to diminish the increased cation permeability observed in hemolytic anemia.

## Discussion

The current model for mechanoelectrical transduction consisting of a force-bearing center seems to be universal for eukaryote cells that sense force[49]. In such model ion channel activity would be suppressed by a stiff platform rich in stomatin-like proteins[50], the plasma membrane[30,31,37], and cytoplasmic and extracellular tethers[13,51,52]. Upon mechanical stimulation, the interaction between the components of this center would lead to ion channel opening, allowing a cell to convert mechanical stimuli into electrical and biochemical signals. The current knowledge of the Piezo channels suggests that they might work in a similar force-bearing center[13]. Importantly, the force-from-lipids principle[53] establishes that the mechanical properties of the membrane control the conformational rearrangements underlying mechanosensitive ion channels gating. As fatty acids are the lipids' building blocks[29] and are known to alter the mechanical properties of the membrane[54,55], our goal was to determine the extent by which the length and unsaturation number of fatty acid-containing lipids modify Piezo1 function. In this study, we present several lines of evidence that support the hypothesis that

distinct fatty acid-containing phospholipids regulate Piezo1 function. First, incorporation of MA in N2A cells and HMVEC inhibits Piezo1 by increasing the mechanical threshold required to activate the channel. Second, decreased Piezo1 activation correlates with the increase in membrane rigidity determined for membranes enriched in MA. Third, long PUFAs supplementation modulates Piezo1 time course of inactivation. Fourth, perfusion of free fatty acids does not modify Piezo1 mechanical response. Fifth, Piezo1 modulation by fatty acids is not affected by disrupting the actin filaments. Sixth, Piezo1 exhibit different gating properties depending on the intrinsic cell fatty acid profile. Our results support the notion that fatty acid-containing phospholipids fine-tune Piezo1 mechanical response.

How do MA-containing membranes inhibit Piezo1 function? MA consumption has been recently associated with potential health benefits, as diets with increased daily MA intake were correlated with attenuating metabolic syndromes related to inflammation like hyperferritinemia and diabetes in dolphins and dogs, respectively[48,56]. Eukaryotic cell membranes are enriched in fatty acids with chain lengths from 18 to 20 carbons carrying more than one double bond[29]; these molecules are tightly controlled by the cell to ensure proper function of membrane proteins. We determined that augmentation of the saturated fatty acid MA in synthetic and native membranes increases structural order and rigidity. Consequently, Piezo1 activation requires twice the displacement to evoke a detectable response in MA-enriched membranes. Other groups have calculated that the change in

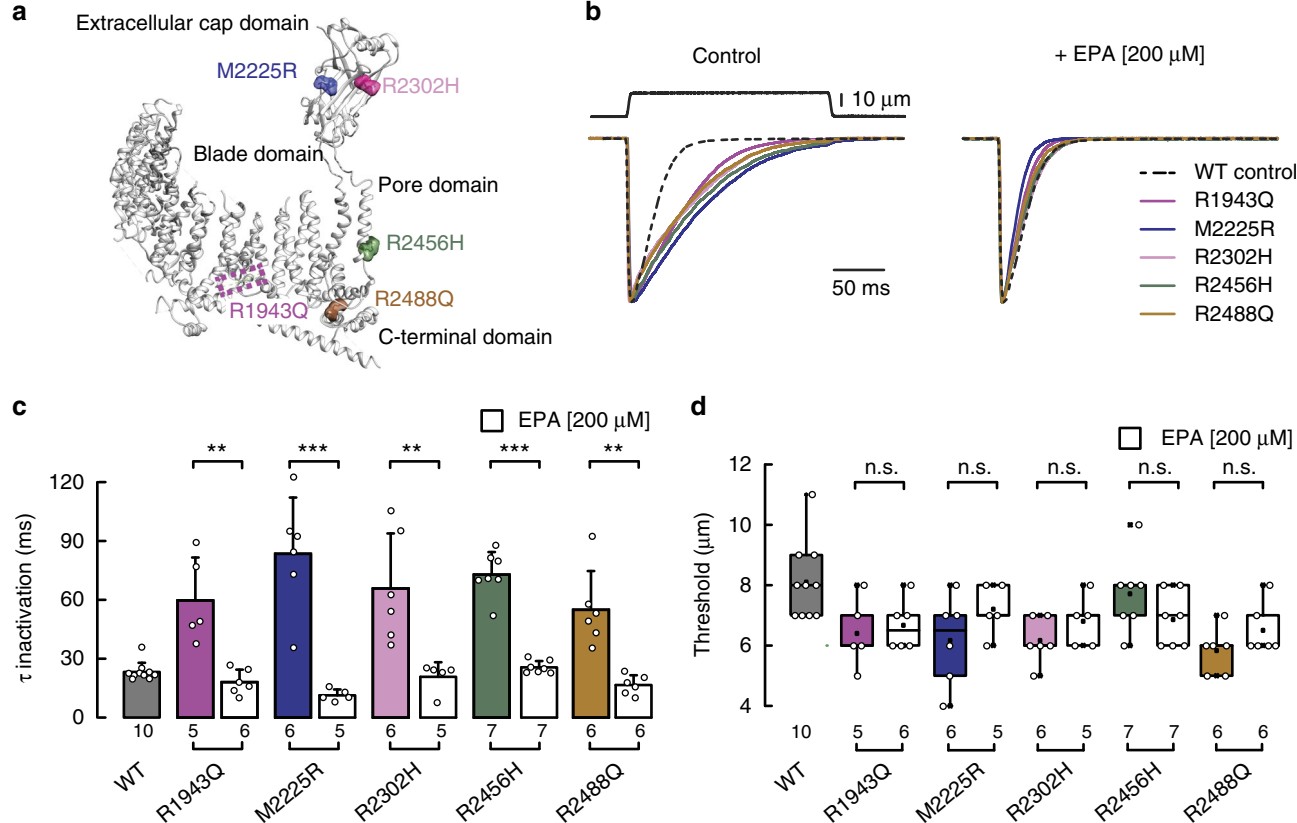

**Fig. 7** EPA supplementation abrogates the phenotype of Piezo1 xerocytosis mutations. **a** Ribbon representation of *Mus musculus* (mm) Piezo1 monomer (PDB ID: 5Z10) highlighting equivalent residues that when mutated cause dehydrated hereditary stomatocytosis in humans. **b** Representative normalized macroscopic currents (at −60 mV) evoked by maximum displacement of N2A cells transfected with human Piezo1 dehydrated hereditary stomatocytosis mutants R1943Q, M2225R, R2302H, R2456, and R2488Q with and without EPA supplementation (left and right, respectively). Human Piezo1 wild type (WT) without supplementation is shown for comparison. **c** Piezo1 time constants of inactivation elicited by maximum displacement of dehydrated hereditary stomatocytosis mutants R1943Q, M2225R, R2302H, R2456H, and R2488Q with and without EPA supplementation. Human Piezo1 WT without supplementation is shown for comparison. Bars are mean ± SD. Unpaired *t*-test with Welch correction, except for R2302H in which Mann–Whitney test was used. **d** Boxplots show the mean, median, and the 75th to 25th percentiles of the displacement threshold required to elicit currents of Piezo1 dehydrated hereditary stomatocytosis mutants R1943Q, M2225R, R2302H, R2456H, and R2488Q with and without EPA supplementation. Human Piezo1 WT without supplementation is shown for comparison. Unpaired *t*-test, except for R2488Q in which Mann–Whitney test was used. Asterisks indicate values significantly different from control (***$p < 0.001$ and **$p < 0.01$) and n.s. indicates not significantly different from the control. $n$ is denoted above the *x*-axes

Piezo1 in-plane area upon opening is ~15 nm$^2$ and that the tension required to activate it is ~5 mN m$^{-1}$ [23]. We hypothesize that the change in Piezo1 in-plane area becomes restricted during mechanical gating in the context of MA-enriched membranes, likely shifting the tension needed to activate the channel to magnitudes closer to the ones reported for the mechanosensitive channel of large conductance (MscL, ~12 mN m$^{-1}$)[57]. Piezo1 structures revealed a colossal trimer, with 38 transmembrane segments per monomer, shaped like a propeller with three blades organized around the permeation pathway[22,46,58–60]. Mutations in the long intracellular helix termed the beam reduced 47% of Piezo1 mechanical currents[46]; the beam with two structure-like hinges (latch and clasp) are thought to transmit membrane deformation from the transmembrane blades to the permeation pathway[60]. Rigid membranes containing MA might impair proper force transmission from the blades to the pore impairing Piezo1 mechanoresponse.

From other ion channel families, it is known that inactivation is due to the ion channel's intrinsic properties (e.g. K$^+$ channels C-type inactivation[61]) and/or channel-accessory subunit interactions (e.g. the K$^+$ channel Kv7.1 with minK[62]). The Piezo family is no exception, as it has been shown that Piezo1 inactivation is determined by the C-terminal extracellular domain

(CED) and the inner pore helix[42,63]. In the case of Piezo2, inactivation is modulated by a region located near the intracellular latch[64]. Importantly, Wu and collaborators demonstrated that swapping the CED between Piezo1 and Piezo2 exchanges their time courses of inactivation[42]. Here, we demonstrated that inactivation is also regulated by the presence of long PUFAs contained in membrane phospholipids. It is possible that PUFAs affect inactivation by modulating the allosteric coupling between the CED and the inner pore helix. Alternatively, PUFAs can alter the interaction between Piezo1 and other proteins. Indeed, it has been reported that Piezo1 inactivation is also modulated by TMEM150C, a transmembrane protein that belongs to a family that regulates phosphatidylinositol 4-phosphate synthesis[65]. In this scenario, it would be possible that plasma membranes enriched with long PUFAs alter Piezo1 modulation by TMEM150C or other proteins.

Our data shows that AA and EPA enhance Piezo1 inactivation, whereas DHA reduces it; this was unexpected because they all decrease plasma membrane rigidity and bending stiffness. We propose that the additional unsaturation of DHA, that is closest to the carboxylic acid group, has a larger structural effect at the membrane/channel interface, likely reducing the cohesion between neighboring lipids (reflected in the lack of cooperativity

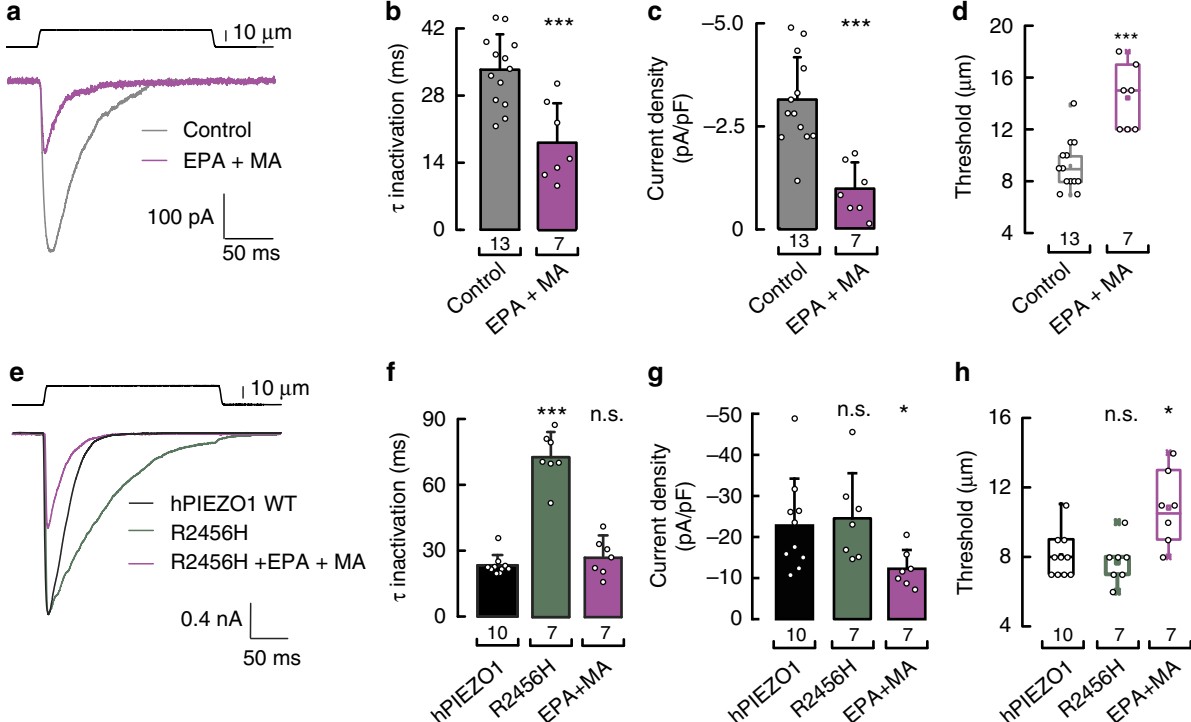

**Fig. 8** EPA and MA have distinct and synergistic effects on Piezo1 gating. **a** Representative macroscopic current (at −60 mV) evoked by maximum displacement of N2A cells treated with and without a mixture of EPA and MA (200 and 100 μM, respectively). **b** Piezo1 time constants of inactivation elicited by maximum displacement of N2A cells treated with and without a mixture of EPA and MA. Bars are mean ± SD. One-way ANOVA and Bonferroni test. **c** Current densities elicited by maximum displacement of N2A cells treated with and without a mixture of EPA and MA. Bars are mean ± SD. Unpaired *t*-test. **d** Boxplots show the mean, median, and the 75th to 25th percentiles of the displacement threshold required to elicit Piezo1 currents of N2A cells treated with and without a mixture of EPA and MA. Unpaired *t*-test. **e** Representative normalized macroscopic currents (at −60 mV) evoked by maximum displacement of N2A cells transfected with human Piezo1 wild type (WT) and R2456H, with and without a mixture of MA and EPA supplementation (200 and 100 μM, respectively). **f** Piezo1 time constants of inactivation elicited by maximum displacement of N2A cells transfected with human Piezo1 WT and R2456H, with and without a mixture of MA and EPA supplementation. Bars are mean ± SD. Kruskal–Wallis and Dunn's multiple comparisons test.
**g** Current densities elicited by maximum displacement of N2A cells transfected with human Piezo1 WT and R2456H, with and without a mixture of MA and EPA supplementation. Bars are mean ± SD. Kruskal–Wallis and Dunn's multiple comparisons test. **h** Boxplots show the mean, median, and the 75th to 25th percentiles of the displacement threshold required to elicit Piezo1 currents of N2A cells transfected with human Piezo1 WT and R2456H, with and without a mixture of MA and EPA supplementation. Kruskal–Wallis and Dunn's multiple comparisons test. Asterisks indicate values significantly different from control (***$p < 0.001$ and *$p < 0.05$) and n.s. indicates not significantly different from the control. *n* is denoted above the *x*-axes

of the DHA's thermogram) and the transmembrane domain to stabilize the open state. Alternatively, the antagonizing effect of PUFAs could be attributed to the ability of DHA to further alter membrane organization. Indeed, it has been shown that DHA, but not EPA, has the ability to stabilize raft microdomains[35,36,66]. Nevertheless, the discrepancy in inactivation observed between AA and EPA with DHA remains unexplained; future in-depth studies are required to understand this effect.

Why Piezo1 activity changes between cell types? Piezo1 displays distinct inactivation profiles in N2A cells and HMVEC; our results supplementing N2A cells with the ω-6 fatty acids precursor linoleic acid supports the idea that inactivation is modulated by the cellular fatty acid distribution. A recent work has shown that Piezo1 displays a non-typical slowly inactivating currents in mouse embryonic stem cells that becomes faster as the cell differentiates into motor neurons[67]. Importantly, there is evidence that de novo lipogenesis and changes in fatty acid synthesis, lipid metabolism, and membrane fluidity play an important role during differentiation[66,68–70]. For instance, differentiation of human mesenchymal stem cells into osteoblasts or adipocytes results in remodeling of the plasma membrane, yielding cell-specific membrane compositions with distinct biophysical properties[66]. Therefore, it remains to be determined

whether the plasma membrane composition (e.g., fatty acids distribution) contributes to the different inactivation kinetics displayed by Piezo1 during cell differentiation. Taken together, the inherent cell lipid profile and dynamic changes in the fatty acid metabolism could add a level of regulation to fine-tune Piezo1 response to mechanical stimuli.

Several studies have shown that gain-of-function mutations in Piezo1 are linked to dehydrated hereditary stomatocytosis[16–19], in which erythrocytes are abnormally shaped due to cation imbalance[19]. Noteworthy, mutations R1942Q, M2224R, R2301H, R2456H, and R2487Q are scattered throughout the channel (Fig. 7a); suggesting that inactivation is modulated by various regions of the protein. Our findings demonstrate that EPA supplementation abrogates the slowly inactivating phenotype of these Piezo1 mutations. EPA-enriched membranes likely exert a ubiquitous effect on the protein rather than modulating inactivation through specific binding sites. Indeed, our findings are further supported by the fact that perfusion of free EPA does not change Piezo1 activity. When looking for alternative supplementations protocols to counterbalance these mutations deleterious effects, we found that combining MA and EPA decreases channel currents and the time constant of inactivation of both wild type and the mutation R2456H (Fig. 8). This result supports the notion

that fatty acids have distinct and synergistic effects on Piezo1 gating. Interestingly, ω-3 fatty acids were used in medical trials as an effective therapy for ameliorating the symptoms associated with sickle cell anemia[71–73]. Hence, we envision that combining dietary fatty acids to modulate different Piezo1 gating modalities could be an effective therapeutic strategy to diminish the effect of hereditary mutations that alter erythrocytes' cation balance.

## Methods

**Cell culture and electrophysiology**. Mouse neuro-2a (N2A; catalog number CCL-131) and human embryonic kidney (HEK-293; catalog number CRL-1573) cell lines were obtained from the American Type Culture Collection (ATCC®) and HMVEC (catalog number ACBRI 539) from neonatal human dermal donor isolates at passage 4 from Cell Systems Corporation. N2A, HEK-293 and HMVEC were cultured according to the manufacturer's protocol and used up to the 25th, 30th, and 7th passage, respectively. Prior to electrophysiological measurements, N2As, HEK-293, and HMVEC were supplemented overnight (≈18 h) with 100 μM (unless otherwise noted) pentadecanoic acid (PDA), heptadecanoic acid (MA), linoleic acid (LA), arachidonic acid (AA), EPA, or docosahexaenoic acid (DHA)[31]. For fatty acid accumulation assays, N2As were supplemented with 10 μM MA or EPA every 24 h for 5 days; and HMVECs supplemented with 50 μM EPA every 24 h for 3 days and used the following day. Fatty acid vials were obtained from Nu-Chek Prep, Inc and were freshly opened prior supplementing the cells media.

For whole-cell recordings, the bath solution contained 140 mM NaCl, 6 mM KCl, 2 mM CaCl$_2$, 1 mM MgCl$_2$, 10 mM glucose, and 10 mM HEPES (pH 7.4). The pipette solution contained 140 mM CsCl, 5 mM EGTA, 1 mM CaCl$_2$, 1 mM MgCl$_2$, and 10 mM HEPES (pH 7.2). Fatty acids and channel agonists and antagonists perfused during the experiments were dissolved in the bath solution: Yoda1 (Tocris Bioscience), GSK-417651A (FOCUS Biomolecules), HC-030031, HC-067047, and GdCl$_3$ (Sigma-Aldrich), and L-GsMTx-4 (Abcam). Pipettes were made out of borosilicate glass (Sutter Instruments) and were fire-polished before use until a resistance between 3 and 5 MΩ was reached. Macroscopic currents were recorded in the whole-cell patch clamp configuration at a constant voltage (−60 mV, unless otherwise noted), sampled at 100 kHz and low-pass filtered at 10 kHz using a MultiClamp 700B amplifier and Clampex (Molecular Devices, LLC). Leak currents before mechanical stimulations were subtracted off-line from the current traces and data were digitally filtered at 2 kHz with ClampFit (Molecular Devices, LLC). Recordings with leak currents >200 pA, with access resistance >10 MΩ, and cells which giga-seals did not withstand at least 6 consecutive steps of mechanical stimulation were excluded from analyses. The time constant of inactivation τ was obtained by fitting a single exponential function, Eq. (1) between the peak value of the current and the end of the stimulus:

$$f_{(t)} = \sum_{i=1}^{n} A_i * e^{-t/\tau_i} + C \qquad (1)$$

where $A$ = amplitude; $\tau$ = time constant; and the constant y-offset $C$ for each component $i$.

**Mechanical stimulation**. N2As, HEK-293, and HMVEC were mechanically stimulated with a heat-polished blunt glass pipette (3–4 μm) driven by a piezo servo controller (E625, Physik Instrumente). The blunt pipette was mounted on a micromanipulator at an ~45° angle and positioned 3–4 μm above from the cells without indenting them. Displacement measurements were obtained with a square-pulse protocol consisting of 1-μm incremental indentation steps, each lasting 200 ms with a 2-ms ramp in 10-s intervals. The threshold of mechano-activated currents for each experiment was defined as the indentation step that evoked the first current deflection from the baseline. For I–V relationships, the holding potential was changed for 5 s before mechanical stimulation. Only cells that did not detach throughout stimulation protocols were included in the analysis. The piezo servo controller was automated using a MultiClamp 700B amplifier through Clampex (Molecular Devices, LLC).

**Liquid chromatography–mass spectrometry**. Control and fatty acid-treated N2A cells and HMVECs were cultured according to manufacturer's protocols up to 2–4 million cells. Cells were rinsed with PBS three times and frozen in liquid N$_2$. Total and free fatty acids were extracted and quantified at the Lipidomics Core Facility at Wayne State University. Membrane (i.e., esterified) fatty acids were determined by subtracting free from total fatty acids and normalized by the number of cells in the culture. The Source Data underlying Figs. 1d, 2e, 4a, 5e, g, 6b, and Supplementary Figs. 1g, 2c, 7a are provided as a Source Data file: https://doi.org/10.6084/m9.figshare.7710140.

**Differential scanning calorimetry**. DSC experiments were performed as previously described[31]. DPPC (4 mM) was dissolved in ethanol and mixed with specified fatty acids. Lipid mixtures containing 5 mol% of individual PUFAs were evaporated. Lipid mixtures were hydrated with 10 mM HEPES (pH 7) at ≈50 °C. To yield multilamellar vesicles, the suspension was vortexed and incubated for 1 h at ≈50 °C. Samples were degassed at 635 mmHg for 10 min at 3 °C and equilibrated

for 5 min at 4 °C prior to DSC experiments. Measurements were performed on a NanoDSC microcalorimeter (TA Instruments). Thermograms were recorded at a constant scan rate of 1 °C min$^{-1}$ (between 10 and 60 °C), and a pressure of 3 atm, and were analyzed with Nano Analyze.

**Atomic force microscopy**. N2A cells were supplemented with 100 μM fatty acids overnight (≈18 h) prior to measurements. To disrupt the cytoskeleton, control and fatty acid supplemented cells were incubated with 1 μM latrunculin A (Cayman Chemicals) for 1 h, as established elsewhere[40]. AFM experiments were carried out as previously described[31,74] with a Bruker BioScope Resolve™ atomic force microscope using the contact mode. In brief, AFM cantilevers MLCT-Bio-DC A probe (Bruker) were coated for 1 h with 1 mg ml$^{-1}$ peanut lectin in PBS (pH 7.4) and calibrated per the thermal noise method[75] with spring constants of 55–65 pN nm$^{-1}$. Membrane tethers were pulled at 40 μm s$^{-1}$ from N2A cells at 25 °C. To optimize the number of bonds between the peanut-lectin coated cantilever and the membrane, the contact force was 3.26 nN while the contact time was 750 ms. Individual ramp scans were used to generate force–distance curves displaying unbinding events. For cell-cantilever contacts that yielded multiple-tether events, forces were characterized by the length of the last observed ruptured tether[39]; membrane tethers refer to the length of ruptured events (Supplementary Figure 3a). Data were obtained from at least three tissue culture plates (≥25 cells per plate), acquired and analyzed with NanoScope (Bruker Corporation) and presented using estimation plots[76]. Data points >200 pN were excluded.

**Calcium imaging**. HMVEC were loaded with Fluo-4 AM (1 mM; Invitrogen) per the manufacturer's protocols. Images were acquired and analyzed with CellSens Dimensions (Olympus Corp.). The solution contained 140 mM NaCl, 2 mM CaCl$_2$, 1 mM MgCl$_2$, 10 mM HEPES, 10 mM glucose, and 20 mM mannitol (320 mOsm) (pH 7.3). Increases in fluorescence were achieved by perfusing 2 μM of Yoda1 (Tocris Bioscience) and washed with control solution. Cells with a large baseline fluorescence signal (>20 arbitrary units) were excluded from the analysis.

**siRNA-mediated knockdown and q-RT-PCR**. HMVEC were transfected with Lipofectamine® RNAiMAX Transfection Reagent (Thermo Fisher Scientific) according the manufacturer's protocols[1]. The siRNA concentration was 20 nM for Piezo1 and silencer negative control (Thermo Fisher Scientific). The transfection was done with free-antibiotic media and after 6 h of transfection, the medium was replaced with a fresh one containing antibiotics. For electrophysiology experiments, cells were also co-transfected with siGLO Green Transfection Indicator (Dharmacon™). The targeted sequences of siRNAs directed against human Piezo1 RNA (ThermoFisher Scientific) was: 5′-AAGAAGAUCGUCAAGUACG-3′. The negative control (scrambled siRNA) was purchased from Ambion, Inc.

RNAs were isolated from HMVEC using RNeasy® Mini Kit (QIAGEN). cDNA was generated with iScript™ Reverse Transcription Supermix for RT-qPCR (Bio-Rad). qPCR triplicate reactions were run in a CFX Connect™ Real-Time PCR detection system with SsoAdvanced™ Universal SYBR® Green supermix (Bio-Rad) according to the manufacturer's instructions. Data acquisition and analysis was done using Bio-Rad CFX Maestro 1.0 (Bio-Rad Laboratories). Normalization was done using ΔΔCq with GAPDH as the reference gene. Standard curves were generated from specific templates for each PrimePCR™ assay mentioned below. Primers were purchased from Bio-Rad for Human GAPDH (Unique Assay ID: qHsaCED0038674) and hPiezo1 (Unique Assay ID: qHsaCID0012344).

**N2A and HEK-293 cells transfection**. N2A and HEK-293 cells were transfected with 1 μg of human PIEZO1 wild type and mutants (R1943Q, M2225R, R2302H, R2456H, and R2488Q), cloned in pIRES2-EGFP[45], using Lipofectamine 2000 (Thermo Fisher Scientific) according to the manufacturer's instructions and recorded 48 h later. Overexpression of Piezo1 wild-type and gain-of-function mutants elicited currents that were 10-times higher than the Piezo1 native ones (Supplementary Figure 8a). Fatty acids were supplemented 18–24 h prior recording.

**Data analysis**. Results were expressed as means ± SD (unless otherwise noted). Box plots depict a range between the 25th and 75th percentiles, mean, median, and outliers with a 1.5 coefficient. Data were plotted using OriginPro (from OriginLab) and QtiPlot (from Ion Vasilief). Sigmoidal fitting was done using OriginPro with the following Boltzmann function:

$$f_{(x)} = A_2 + \frac{A_1 - A_2}{1 + e^{((X-X_o)/dX)}} \qquad (2)$$

where $A_2$ = final value; $A_1$ = initial value; $X_o$ = center; and $dX$ = time constant.

Statistical analyses were performed using GraphPad Instat 3 software. One-way ANOVA with Bonferroni test, repeated measures analysis of variance with Bonferroni test, two-tailed paired and unpaired $t$-test were used to evaluate statistical significance between samples. When the SD's of the tested groups were significantly different, and/or the data were not sampled from a normal distribution, the statistical significances were evaluated with Kruskal–Wallis with Dunn's test, Friedman's test with Dunn's test, or Mann–Whitney test. Individual tests are described on each of the Figure legends.

**Reporting summary**. Further information on experimental design is available in the Nature Research Reporting Summary linked to this article.

## Data availability

Data supporting the findings of this manuscript are available from the corresponding author upon reasonable request. A reporting summary for this Article is available as a Supplementary Information file. The Source Data underlying Figs. 1b–g, 2b–f, 3b–d, 4a, 4c–e, 5b–f, and 6b, 6d, 7c, d, 8b–d, 8f–h, and Supplementary Figs. 1a, b, 1d–g, 2b–d, 3b, 3d–f, 4a, b, 5a, 5f, 6a, b, 7a, 8a, 8c–e, 8g are provided as a Source Data file.

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

## Acknowledgements

We thank Dr. J.C. Ruiz-Suárez for providing access to the DSC instrument; Dr. P.A Gottlieb for providing human Piezo1 R2456H in pIRES2-EGFP; and Dr. A. Chesler for technical advice and critically reading the manuscript; Dr. C.M. Waters and Dr. R.C. Foehring for critically reading the manuscript; and members of the Cordero and Vásquez laboratories for technical support. We used the lipidomics Core Facility at Wayne State University (NIH S10RR027926). This work was supported by the National Institutes of Health (R01GM125629 to J.F.C.-M.; and R01CA210192, R01CA206069, and R01CA204552 to S.C.C.), the United States-Israel Binational Science Foundation (2015221 to V.V.), and the American Heart Association (16SDG26700010 to V.V. and 15SDG25700146 to J.F.C.-M.).

## Author contributions

Conceptualization: V.V.; methodology: V.V., J.F.C.-M. and L.O.R.; formal analysis: V.V. and L.O.R.; investigation: L.O.R., F.J.S.-V., A.E.M. and A.D.M.-D.; resources: V.V., J.F.C.-M. and S.C.C, writing – original draft preparation: V.V.; writing – review & editing: V.V. and J.F.C.-M.; supervision: V.V. and J.F.C.-M.; project administration: V.V.; funding acquisition: V.V. and J.F.C.-M.

## Additional information

**Competing interests:** The authors declare no competing interests.

