## [Peer Review File · Nature Communications]

Reviewers' Comments:

Reviewer #1:

Remarks to the Author:

The manuscript entitled "DIETARY FATTY ACIDS FINE-TUNE PIEZO1 MECHANICAL RESPONSE" discusses the influence of fatty acids on the function of the mechanosensitive protein Piezo1. The authors find that margaric acid increases the activation threshold for Piezo channels while cocosaheaxaenoic acid increases the inactivation kinetics keeping the receptor open for longer. The authors then nicely apply their findings and use these fatty acids to partially reverse symptoms related to Piezo gain of function and inactivation mutations which could be used in the future as treatment of red blood cell disorders.

The manuscript is well written and easy to comprehend. The results are novel and noteworthy. However to make it accessible for the broad readership of nature communications and to convince this reviewer, a few points need to be addressed. In general, the manuscript would be much improved if more attention would be given to statistical tests. This reviewer is also wondering by what mechanism the fatty acids modulate piezo. Do the authors have any idea if this is a direct or indirect modulation, i.e. are the lipids directly associate with the Piezo protein or if it's the membrane properties that indirectly affect Piezo? can you pull down those fatty acids in a CoIP assay? Is the observed effect specific to Piezo or more general? i.e. are other mechanosensitive proteins (e.g. Piezo2 or TRPV4) also modulated by addition of MA? I would like to see an extra experiment that resolves one of the the questions on either direct vs indirect effect or specificity of MA. for example the authors could try overexpression of another mechaonsosensitive protein in the same experimental context and testing effect of MA. The authors make a lot of hypotheses about mechanisms in the discussion but it's a bit like handwaving to me, prompting me to ask for an extra experiment on the mechanism to bolster the discussion section.

Specific remarks:

Introduction:

- Well written and comprehensive.
- Minor comment: to broadly introduce the topic and include other views in the field besides gating by FFL, it would be important to mention gating of mechanosensitive channels via tethers as has been shown to be the base in hair cells, piezo1, and others – see: i) Hu et al Evidence for a protein tether involved in somatic touch. EMBO J. 2010, 29 (4), 855–67. ii) Assad, J. A.; Shepherd, G. M.; Corey, D. P. Tip-link integrity and mechanical transduction in vertebrate hair cells. Neuron 1991, 7, 985–994. and iii) Gaub et al. Mechanical Stimulation of Piezo1 Receptors Depends on Extracellular Matrix Proteins and Directionality of Force, Nano Lett. 2017, 17, 2064–2072)

Results:

- I don't understand the link between these experiments and prevalence of fatty acids in dairy products. How is it of importance this Piezo mechanistic study if the fatty acids Please elaborate
- "IC50 = 28.3 +/- 3.4 μM" please indicate what these numbers are. Mean +/- SEM or SD?
- " We consistently required high displacement steps (> 16 μm)" how are displacements of 16um possible? N2A cells I assume are only about 10um in height so how would you displace them by 16um? I have issues understanding how cells, in general, and how the giga ohm seals of the patch in particular, can survive this type of stimulation. Please resolve how this distance was measured.
- "it is tempting to suggest that the additional unsaturation of the DHA, that is closest to the carboxylic acid group, has a larger structural effect at the membrane/channel interface." – this sentence should be moved to discussion section.
- "We found that the mean tether force for control N2A cells (59 ± 19.4 pN; Fig. 2F) is similar to the ones reported for other cultured cells and neurons. As expected, we observed that MA treatment significantly increased the mean tether force of N2A cells (103.5 ± 38.3 pN) as compared to untreated ones (Fig. 2F) " The three distributions of mean tether forces in Fig 2F are

all overlapping. Therefore, find it surprising that the mean tether values are so dramatically different. Also I find it surprising that the data points for MA are so widely distributed (50-200pN) whereas control and PDA cluster much more closely together. Why is this the case? I propose two changes: i) More data points for MA condition, and ii) statistical test for significance

- "treatment with PDA – another saturated fatty acid – did not alter the elastic properties of N2A plasma membranes (48.2 +/-16.7; Fig. 2F)". how can the authors be certain that PDA actually integrated into the membranes? For MA the uptake into cellular membrane is shown (Fig 1F) and for DHA as well (Fig2C). it would be important to check that the lack of effect of PDA does not stem from a lack of uptake.

- "Furthermore, we determined that Piezo1 is functional in HMVEC when measuring an increase in the fluorescence signal after addition of Yoda1 to cells loaded with a Ca²⁺-sensitive dye (Fluo-4 AM)." I have a few issues with this experiment and I am not convinced by the data as it is presented in the manuscript. Specifically:

i) Why are the traces so symmetric in terms of signal rise and decay. Normal calcium responses that are caused by channel opening show a sharp signal increase and an exponential decay (provide example refs here). These traces look rather symmetric.

ii) Has all the data been obtained from only one experiment?

iii) Why are the responses so short? Others have shown calcium responses lasting seconds – minutes before returning to baseline (see for example Syeda et al. Elife. 2015 May 22;4. doi: 10.7554/eLife.07369. and Lacroix et al. Nature Communications volume 9, Article number: 2029 (2018)). These signals are very short lasting only a few hundred ms. Please explain.

iv) Why do you see activation by Yoda1 alone in absence of force? I was under the impression that yoda1 acts as an agonist to Piezo1 only in presence of mechanical stimulation (Syeda et al. Elife. 2015 May 22;4. doi: 10.7554/eLife.07369)

- "hence, in this cellular context Piezo1 gating is mainly altered by the contribution of individual fatty acids rather than their global distribution (Fig. 4F)" This seems to be in disagreement with the conclusion/header of the next paragraph: "Changing the global fatty acid distribution modifies Piezo1 gating properties"

- " The gain of function mutant R2456H is also characterized by a leftward shift of the activation pressure and displacement threshold when compared to the wild-type channel (Fig. 6E)" - I don't see the difference. All distributions are overlapping. If you added statistics comparing the distributions, like you do in panel F, I would be convinced, but not as its presented here.

- " Notably, EPA and MA reduced the inactivation time and current densities while increasing the displacement threshold of the R2456H mutant (Fig. 6F) to levels reminiscent of the wild-type channel" – I don't see any evidence for this claim regarding force thresholds. If I understand correctly, piezo1 WT thresholds (Fig 6E, black) and R2456H +EPA + MA (Fig 6F, purple) should be the same which they are clearly not. Please provide a side by side comparison with statistics to convince me otherwise.

Methods:

AFM: The resolve AFM is typically used for imaging and not for force distance curves. Therefore, please describe the full protocol that was used for AFM experiments. Was force volume or imaging mode used to obtain the data? What contact force was used to pull the tethers? For the analysis of force curves, what criteria were used to determine what are tethers (i.e. slope of FD curves or length of rupture events) ? Please elaborate on this.

Figures:

Fig1 G, Fig 2 D, Fig 3 A, Fig 4 G, Fig 5 B – Run parts of the whole analysis, e.g chi square test
Fig 1E - Where are the mean values for 0 and 1uM MA? Do both fall on the lower edge of the boxplot?

Fig 2E – use different colors, these are hard to separate

Fig 2F – statistics needed for mean tether force comparison

Fig 2F – Force distance curves. Label top with "control" and "+MA" on the traces to make it easier

to read. bottom left: show zoomed image as inlay for the rupture event in both of the black traces. its very hard to see in the small magnification

Fig 6A – I don't understand why this cartoon is needed. How does this help illustrate the experiments done in figure 6?

Fig 6E – statistics needed for threshold force comparison

Reviewer #2:

Remarks to the Author:

The manuscript by Romero et al studies the effect of dietary fatty supplementation on Piezo1 mechanically activated currents. The key findings are the following. In N2A cell, supplementing the cell culture media overnight with the fully saturated margaric acid (MA) results in the almost complete disappearance of Piezo1 currents in response to the maximal stimulus used in their assay. The Piezo1 agonist yoda1 recovers the currents in MA treated cell, indicating that the channels were present in the membrane. The poly-unsaturated fatty acids AA, EPA and DHA do not affect current amplitudes, but they alter inactivation kinetics, AA and EPA speeds up inactivation, while DHA slows it down.

The authors also identify endogenous Piezo1 currents in human microvascular endothelial cells (HMVEC). In these cells, mechanically activated Piezo1 currents did not display full inactivation, and showed a marked steady state phase. Linoleic acid supplementation of N2A cells rendered Piezo1 currents more similar to that in HMVEC cells, showing incomplete inactivation. In HMVEC cells MA induced a strong reduction in current density, and DHA almost completely removed inactivation similar to the effects observed in N2A cells. EPA and AA on the other hand failed to significantly increase the speed and extent of inactivation.

The authors also studied a slowly inactivating Piezo1 mutant that causes xerocytosis, and found that when expressed in N2A cells, supplementation of EPA increased the speed of inactivation of this mutant similar to that found in wild type Piezo1.

Overall, the manuscript presents novel data demonstrating that dietary fatty acid supplementation can significantly alter both the amplitude and the inactivation kinetics of Piezo1 currents.

Comments:

1. The authors claim that the effects on Piezo1 are mediated by altering the physical properties of the plasma membrane. If that is the case it would be expected that other ion channels, especially those activated by physical stimuli will also be affected. Control measurements with other ion channels, such as Piezo 2 and some other channels activated by physical stimuli, such as thermoTRP channels would give a sense of the specificity of the effect.

2. The authors claim in the abstract: "A combination of dietary fatty acids might be a viable treatment for attenuating the symptoms associated with red blood cell disorders." I believe this statement is premature for several reasons. First, the authors show that EPA speeds up Piezo1 inactivation in N2A cells, but not in HMVEC cells. What if the fatty acid composition of red blood cells is such that EPA will have no effect on Piezo1 inactivation? I would at least try a third cell type, expressing heterologous or native Piezo1 to see if the effect is generally applicable in other cell types. See also points 3 and 4.

3. Free fatty acids in the blood plasma bind to proteins, mainly albumin. Protein concentration in the plasma, where red blood cells are located, is ~70g/L, which is much higher than in the interstitium and in standard cell culture media. I am not sure if the same concentrations of fatty acids would have the same effect if they were dissolved in high protein buffer; this is easy to test, and I recommend that the authors address this.

4. The authors only tested one Piezo 1 mutant; I think the data would be stronger if they also examined other Xerocytosis mutants, to see if the effect is generally applicable.

5. The authors show in Figure 1 that acute perfusion of MA has no effect on Piezo1 currents. Similar data should be shown for the unsaturated fatty acids also

6. The authors incubate cells with various fatty acids. From the manuscript it is not entirely clear if these incorporate in the plasma membrane as free fatty acids, or they are incorporated in phospholipids, i.e. covalently bind to a glycerol backbone. This should be clarified and discussed.

Additional minor comments:

Introduction: "Piezo1 seems to be essential in mammals" – this is correct in mice, but the loss of function mutations in humans are compatible with life.

I would put supplemental Figure 2B-D to the main figures, if the authors wish to claim these fatty acids could be therapeutically useful, it is better to show that data that they do not decrease current amplitudes

Reviewer #3:

Remarks to the Author:

This is a rigorously-executed study on the role of dietary fatty acids on Piezo1 activity. The experiments are well designed and the manuscript is clearly written. The authors report novel effects of lipid composition on Piezo1 gating, finding qualitatively different effects of specific fatty acids. Margaric acid (MA) inhibits endogenous Piezo1 in Neuro2A cells, while PUFAs such as DHA and EPA modulate inactivation. Surprisingly, the effects of fatty acid supplementation are strong enough to reverse the effects of a gain-of-function mutation, R2456H, which has been shown to underlie xerocytosis.

Piezo1 has been found to be important for a wide range of physiological processes and disease conditions. An important question in the mechanotransduction field is how the activity of the channel elicits such a broad range of effects. An emerging theme is that the effects of Piezo1 are context dependent. This study introduces the mechanistic insight that plasma membrane lipid composition may be one contributing factor to different effects of Piezo1 in across cellular and organ systems. Thus the finding is of broad significance. The manuscript could be significantly strengthened:

1. Tether-pulling experiments are reported with MA but not with PUFAs. Inclusion of this data is needed to fully support the claim that MA and PUFAs have opposite effects on the membrane mechanics which influences Piezo1 gating. Minor: panels on tether-pulling experiments in Fig. 2F are not labeled as to which is control and which MA-treated.

2. Does the change in mean tether force arise from changes in mechanics of the lipid membrane or from changes in the connection between the membrane and the cytoskeleton? Tether-pulling assays on nascent membrane blebs or after treatment with cytoskeletal disrupting agents would be insightful in determining how the modifying membrane composition affects Piezo1 gating.

3. Activation time course data throughout the paper: Since the channel activates at sub-millisecond timescales which are not resolved by the standard poking assay, presenting activation tau data in different conditions is not informative. Hence, these data should be removed. If the

authors have a stimulus probe that stimulates the cell fast enough to resolve Piezo1 activation kinetics, further evidence should be presented demonstrating this.

4. Fig. S3: GdCl₃ concentration reported (30 mM) is very high. Typically a 10-fold lower (30 Micromolar) GdCl₃ is used. The authors should clarify the concentration used, and perform experiments at micromolar concentrations if not done. Statistical analysis is appropriate.

Minor points:

1. References cited:

- Introduction, para 1, line 4: Also add new paper on Piezo1 in blood pressure regulation by Zeng et al. Science 2018.
- Introduction, para 2, line 1 (and Results, para 1, line 4): "While Piezo1 it is widely agreed upon that Piezo1 is activated by membrane tension...", also include Syeda et al. Cell Reports 2016.
- Section "Piezo1 mediated indentation-activated currents in human microvascular endothelial cells", line 2: "...where it has been shown to gate in response to shear stress" also include Ranade et al PNAS 2014

2. Discussion paragraph 2: units of area change should be in nm²

3. Legend of Fig. 6F, line 1: Legend includes Piezo1 WT but there is no WT data in the panel

4. Results section "Fatty acids alter Piezo1 gating by changing the physical properties of the plasma membrane", sentence: "Our results suggest that although saturated fatty acids (like MA) control the extent of Piezo1 activation, once Piezo1 is open, long PUFAs determine the degree of its inactivation.": Control and treated cells in Fig. 2A show a similar extent of inactivation, but difference in inactivation time courses, thus, the "degree of inactivation" should be revised to "time course of inactivation"

It would helpful to have page numbers and line numbers in the manuscript.

We thank the reviewers for the time invested in our manuscript and their enthusiasm for our work. Thanks to their comments and suggestions, our work has improved with new results and text additions, that we think make our conclusions stronger. Please, find below a point-by-point response to the specific issues raised by the reviewers. All of the additions and changes are highlighted in yellow throughout the manuscript file.

Reviewer 1:

1- *“The manuscript would be much improved if more attention would be given to statistical tests.”*

We agree with the reviewer and have included more details on the statistical tests performed throughout the manuscript, including the abbreviation n.s. (not significant) over the corresponding results.

2- *“This reviewer is also wondering by what mechanism the fatty acids modulate piezo. (a) Do the authors have any idea if this is a direct or indirect modulation, i.e. are the lipids directly associate with the Piezo protein or if it’s the membrane properties that indirectly affect Piezo? (b) can you pull down those fatty acids in a CoIP assay?”*

[Redacted]

The authors make a lot of hypotheses about mechanisms in the discussion but it’s a bit like handwaving to me, prompting me to ask for an extra experiment on the mechanism to bolster the discussion section.”

(a) We understand the concern of the reviewer and now have included in the revised manuscript experiments that shine a light on this issue. Our data suggests that the effect of fatty acids is due to membrane remodeling rather than an acute effect (i.e., binding); as perfusion of free saturated and unsaturated fatty acids to cells while poking under the whole-cell configuration do not change Piezo1 currents (Fig. 1g, Fig. 2f, Supplementary Figure 1h, and Supplementary Figure 2d). Final determination of fatty acids interaction will come from site-directed mutagenesis and structural determination of protein and fatty acids complexes, which we believe is out of the scope of the present manuscript.

(b) The reviewer makes good points. Unfortunately, we have not found antibodies that specifically bind to Piezo1; we have tried primary antibodies (from Thermo Fisher Scientific, Alomone Labs, and Proteintech) that are unspecific as they recognize multiple bands. Moreover, to the best of our knowledge, there are no antibodies that recognize specific fatty acids.

[Redacted]

[Redacted]

[Redacted]

3- “Minor comment: to broadly introduce the topic and include other views in the field besides gating by FFL, it would be important to mention gating of mechanosensitive channels via tethers as has been shown to be the case in hair cells, piezo1, and others – see: i) Hu et al Evidence for a protein tether involved in somatic touch. EMBO J. 2010, 29 (4), 855–67. ii) Assad, J. A.; Shepherd, G. M.; Corey, D. P. Tip-link integrity and mechanical transduction in vertebrate hair cells. Neuron 1991, 7, 985–994. and iii) Gaub et al. Mechanical Stimulation of Piezo1 Receptors Depends on Extracellular Matrix Proteins and Directionality of Force, Nano Lett. 2017, 17, 2064–2072)”

We apologize for the missing information and now have added a section at the beginning of the Discussion section to highlight the contribution of protein tethers to mechanosensitive channels function.

4- “I don’t understand the link between these experiments and prevalence of fatty acids in dairy products. How is it of importance this Piezo mechanistic study if the fatty acids Please elaborate”

We apologize for the misunderstanding. We did not want to establish a link between the presence of margaric and pentadecanoic acids (MA and PDA, respectively) in dairy products and Piezo1 function. Instead, we wanted to provide a context for readers that are not familiar with these fatty acids, especially because MA and PDA are not as popular as the nowadays-common ω -3 fatty acids (EPA and docosahexaenoic acid, DHA). To avoid this issue, we have restructured the sentence in the Results section that might have been the source of the misinterpretation as follows:

The old sentence reads “Among common dietary fatty acids present in dairy products³⁰, we found basal levels of MA (C17:0) and pentadecanoic acid (PDA; C15:0) in N2A cells”.

The new sentence on page 4, lines 101-102, reads “We found basal levels of the odd-chain saturated fatty acids MA (C17:0) and pentadecanoic acid (PDA; C15:0)”.

5- “IC50 = 28.3 +/- 3.4 μ M” please indicate what these numbers are. Mean +/- SEM or SD?”

Thank you for noticing the oversight. Those numbers refer to the mean \pm SEM and all of the other values on the manuscript are mean \pm SD. We have now indicated this throughout the text.

6- “We consistently required high displacement steps (> 16 μ m)” how are displacements of 16 μ m possible? N2A cells I assume are only about 10 μ m in height so how would you displace them by 16 μ m? I have issues understanding how cells, in general, and how the giga ohm seals of the patch in particular, can survive this type of stimulation. Please resolve how this distance was measured.”

We understand the concern of the reviewer and we would like to clarify it. We delivered the mechanical stimulation with a heat-polished blunt glass pipette driven by a piezo servo controller (a technique first implemented on Piezo channels by the Patapoutian laboratory). We initially gently contact the cell [the typical height for N2A cells ranges from 9-14 μ m (Lulevich et al., 2010)] with the blunt pipette and then back up several steps (each step is equivalent to 1 μ m) before starting the mechanical stimulation. This protocol allows us to be at a safe distance from the bottom of the bath. The movement of the blunt pipette is not perpendicular to the cell, it is rather about 45°, making the displacement required to stimulate the cells larger. In our manuscript, we wanted to emphasize that — using the same protocol—we always require more displacement steps to evoke currents in cells incubated with MA. Importantly, we have included more details in the Methods section under the Mechanical Stimulation subheading. Of note, the displacements reported in our work are within the range of the ones reported by other groups (Anderson et al., 2018; Coste et al., 2010; Glogowska et al.,

2017; Ma et al., 2018; Pathak et al., 2014; Qi et al., 2015; Servin-Vences et al., 2017; Wang et al., 2018; Zhao et al., 2018).

We also worried about cells withstanding mechanical displacement. Indeed, we frequently ran a voltage protocol to evaluate ionic currents after mechanical stimulation and compared the results with unstimulated cells. We observed that mechanical stimulation did not affect their normal voltage responses.

7- “it is tempting to suggest that the additional unsaturation of the DHA, that is closest to the carboxylic acid group, has a larger structural effect at the membrane/channel interface.” – this sentence should be moved to discussion section.”

We agree with the reviewer and have removed this sentence from the Results section. It has been added to the Discussion section.

8- “We found that the mean tether force for control N2A cells (59 ± 19.4 pN; Fig. 2F) is similar to the ones reported for other cultured cells and neurons. As expected, we observed that MA treatment significantly increased the mean tether force of N2A cells (103.5 ± 38.3 pN) as compared to untreated ones (Fig. 2F) “The three distributions of mean tether forces in Fig 2F are all overlapping. Therefore, find it surprising that the mean tether values are so dramatically different. Also I find it surprising that the data points for MA are so widely distributed (50-200pN) whereas control and PDA cluster much more closely together. Why is this the case? I propose two changes: i) More data points for MA condition, and ii) statistical test for significance”

We appreciate the reviewer’s suggestions and have followed the advice closely: i) We have acquired more data points for all the conditions (> 70 membrane tethers, from three independent culture plates). ii) We now have included a statistical test for significance. Furthermore, in the revised version of the manuscript we made an independent figure for differential scanning calorimetry (DSC) and atomic force microscopy (AFM; Fig. 3). We also have included polyunsaturated fatty acids (PUFAs)-treatment in our AFM analysis, as suggested per Reviewer 3. These widespread distributions are commonly found when measuring membrane tether forces with AFM (Caires et al., 2017; Diz-Munoz et al., 2010; Krieg et al., 2008; Muller et al., 2009; Vasquez et al., 2014). We speculate that this might be due to the heterogeneity of the plasma membrane throughout the cell (e.g., lipid domains). Altogether, the DSC and AFM data support our hypothesis that MA increases the stiffness of the plasma membrane whereas AA EPA, and DHA decrease it.

9- “treatment with PDA – another saturated fatty acid – did not alter the elastic properties of N2A plasma membranes (48.2 ± 16.7 ; Fig. 2F)”. how can the authors be certain that PDA actually integrated into the membranes? For MA the uptake into cellular membrane is shown (Fig 1F) and for DHA as well (Fig. 2C). it would be important to check that the lack of effect of PDA does not stem from a lack of uptake.”

We agree with the reviewer’s concern and performed the liquid chromatography-mass spectrometry (LC-MS) analysis of N2A cells treated with pentadecanoic acid (PDA). This new result (now included on Supplementary Figure 1g and in the Results Section of the manuscript) demonstrates that, upon treatment, PDA increases in the plasma membrane. Hence, the lack of effect of PDA supplementation on Piezo1 function does not stem from the lack of uptake, but rather from its inability to modify the mechanical properties of the plasma membrane, as determined with AFM (Fig. 3c).

10- “Furthermore, we determined that Piezo1 is functional in HMVEC when measuring an increase in the fluorescence signal after addition of Yoda1 to cells loaded with a Ca²⁺-sensitive dye (Fluo-4 AM).” I have a few issues with this experiment and I am not convinced by the data as it is presented in the manuscript. Specifically: i) Why are the traces so symmetric in terms of signal rise and decay. Normal calcium responses that are caused by channel opening show a sharp signal increase and an exponential decay (provide example refs here). These traces look rather symmetric. ii) Has all the data been obtained from only one experiment? iii) Why are the responses so short? Others have shown calcium responses lasting seconds – minutes before returning to baseline (see for example Syeda et al. *Elife*. 2015 May 22;4. doi:10.7554/eLife.07369.

and Lacroix et al. Nature Communications volume 9, Article number: 2029 (2018)). These signals are very short lasting only a few hundred ms. Please explain. iv) Why do you see activation by Yoda1 alone in absence of force? I was under the impression that yoda1 acts as an agonist to Piezo1 only in presence of mechanical stimulation (Syeda et al. Elife. 2015 May 22;4. doi: 10.7554/eLife.07369)”.

i) and iii) We apologize to this reviewer since there were several mistakes in the Ca²⁺ imaging panel. First, we mislabeled the x-axis and instead of milliseconds, it should have read seconds (this has been corrected in the revised version). Second, the fluorescence signal decreases because we wash the Yoda1-containing buffer with control solution after the signal rises; the figure now includes bars depicting these solution exchanges. Moreover, when we leave Yoda1 up to the end of the experiment, we also observe a very slow decay (in the order of minutes, please see Supplementary Figure 5a).

ii) The micrograph and traces shown on Fig. 4d (former Fig. 3D) correspond to one representative experiment. We have now included on Fig. 4d a bar graph summarizing all of the data (108 cells, from three independent experiments) comparing perfusion of control, Yoda1-containing solutions, and control to wash.

iv) Similar to our results, several laboratories have shown that Yoda1 mediates Ca²⁺ influx in fluorescence-mediated assays without mechanical stimulation (Cahalan et al., 2015; Lacroix et al., 2018; Syeda et al., 2015; Wang et al., 2016). As the reviewer points out, under patch-clamp conditions Yoda1 is not enough to open Piezo1 in cells without mechanical stimulation. At first glance, these results seem to be counterintuitive, and this apparent discrepancy was first reported by the Patapoutian laboratory in their first Yoda1 paper, which is excerpted below:

“Our electrophysiological experiments in cells suggest that Yoda1 prominently affects the sensitivity and the inactivation kinetics of mechanically induced responses but at best causes a slight mPiezo1 activation in the absence of mechanical stimuli... we also observe prominent Yoda1-dependent calcium responses in cell culture and currents in artificial bilayers in the absence of externally applied forces. Therefore, the discrepancy in various levels of channel activity observed with different assays used here remains unexplained, and future in-depth understanding of mechanism of Yoda1 action on Piezo1 might shed light on these apparently disparate observations.” (Syeda et al., 2015).

Although we have not done any experiments to explain this phenomenon, recent work published by the Pathak laboratory might shine a light on it. Using Total Internal Reflection Fluorescence Microscopy, her group showed that Piezo1 opens (in the absence of external mechanical stimulation) by traction forces produced in the vicinity of force-producing adhesions (Ellefsen et al., 2018; Pathak et al., 2014). This suggests that the Ca²⁺ influx observed with fluorescence-based assays upon Yoda1 perfusion might come from sensitizing Piezo1 channels located in those “hot-spots” to lower internal forces.

11- “hence, in this cellular context Piezo1 gating is mainly altered by the contribution of individual fatty acids rather than their global distribution (Fig. 4F)” This seems to be in disagreement with the conclusion/header of the next paragraph: “Changing the global fatty acid distribution modifies Piezo1 gating properties”

We agree with the reviewer and, to avoid confusing readers, we have removed the sentence from the manuscript.

12- “The gain of function mutant R2456H is also characterized by a leftward shift of the activation pressure and displacement threshold when compared to the wild-type channel (Fig. 6E)” - I don’t see the difference. All distributions are overlapping. If you added statistics comparing the distributions, like you do in panel F, I would be convinced, but not as its presented here.

In our original manuscript, it was not clear that the activation pressures of wild type and R2456H mutant were determined by the Sachs and Gottlieb group (Bae et al., 2013) and our results only pertained to the displacement threshold. Indeed, when measuring the threshold, we did not find

statistical differences between wild type and R2456H when probed with a t-test (we now added n.s. to R2456H, Fig. 7d). To avoid further misunderstanding and in the light of the new xerocytosis mutants added (see response to Reviewer 2) we rewrote the entire section.

13- “Notably, EPA and MA reduced the inactivation time and current densities while increasing the displacement threshold of the R2456H mutant (Fig. 6F) to levels reminiscent of the wild-type channel” – I don’t see any evidence for this claim regarding force thresholds. If I understand correctly, piezo1 WT thresholds (Fig 6E, black) and R2456H +EPA + MA (Fig 6F, purple) should be the same which they are clearly not. Please provide a side by side comparison with statistics to convince me otherwise.

We agree with the reviewer’s comment and have changed the text and figure accordingly. To that end, we eliminated “to levels reminiscent of the wild-type channel” and now show the full comparison between wild type and mutant with and without supplementation (in the revised manuscript version this figure panel is now on Fig. 8e-h). It is important to note that we used 100 μ M MA and 200 μ M EPA to have a strong effect; however, using different fatty acid concentrations might tune channel response to resemble wild-type Piezo1.

14- AFM: The resolve AFM is typically used for imaging and not for force distance curves. Therefore, please describe the full protocol that was used for AFM experiments. a) Was force volume or imaging mode used to obtain the data? b) What contact force was used to pull the tethers? c) For the analysis of force curves, what criteria were used to determine what are tethers (i.e. slope of FD curves or length of rupture events)? Please elaborate on this.

We thank the reviewer for noticing this omission. We have included in the revised version of this manuscript more details in the Methods section. Specifically, we included:

- a) We used Contact Mode to obtain the data.
- b) To optimize the number of bonds between the peanut-lectin coated cantilever and the membrane, the contact force was 3.26 nN while the contact time was 750 ms.
- c) Membrane tethers refer to the length of ruptured events.

15- Fig1 G, Fig 2 D, Fig 3 A, Fig 4 G, Fig 5 B – Run parts of the whole analysis, e.g chi square test

We thank the reviewer for the suggestion. Based on it, we have reorganized the LC-MS data as follows:

- We now report the membrane fatty acid distribution for at least three independent cultures of N2A cells (control and treated with linoleic acid) and HMVEC (Figs. 4a and 6b), including their statistical analysis.
- We also report the linoleic acid content in N2A cells and HMVEC for four independent cultures (Fig. 6a), including their statistical analysis.
- We have removed the fatty acids distribution for treated cells; instead, we now report the incorporation of fatty acids in the membrane after treatment as fold change, as we analyzed the treated cells paired with the untreated ones (Figs. 1d, 2e, 5f, Supplementary Figure. 1g).

16- Fig 1E - Where are the mean values for 0 and 1 μ M MA? Do both fall on the lower edge of the boxplot?

Thank you for noticing this gaffe. We might have lost the means of 0 and 1 μ M MA when exporting the file from Origin to Adobe Illustrator. We have corrected this in the revised version of the manuscript.

17- Fig 2E – use different colors, these are hard to separate.

Thank you for the suggestion. We have changed the line patterns of the differential scanning calorimetry data (now Fig. 3a-b).

18- Fig 2F – statistics needed for mean tether force comparison

We have included the statistical analysis in the new version (Kruskal-Wallis and Dunn's multiple comparisons test; Fig. 3c).

19- Fig 2F – Force distance curves. Label top with “control” and “+MA” on the traces to make it easier to read. bottom left: show zoomed image as inlay for the rupture event in both of the black traces. It is very hard to see in the small magnification

Thank you for noticing this. We have labeled the figure accordingly and it now shows insets of the rupture events (now Supplementary Figure 3a).

20- Fig 6A – I don't understand why this cartoon is needed. How does this help illustrate the experiments done in figure 6?

We have removed the red blood cell cartoon from the figure per reviewer's recommendation.

21- Fig 6E – statistics needed for threshold force comparison.

In the original version of this figure we did not include the abbreviation n.s. Nevertheless, we did perform the statistical analysis and determined that they were not significantly different. To avoid confusion, we have included n.s. throughout the figures as requested.

Reviewer 2:

[Redacted]

2- The authors claim in the abstract: “A combination of dietary fatty acids might be a viable treatment for attenuating the symptoms associated with red blood cell disorders.” I believe this statement is premature for several reasons. First, the authors show that EPA speeds up Piezo1 inactivation in N2A cells, but not in HMVEC cells. What if the fatty acid composition of red blood cells is such that EPA will have no effect on Piezo1 inactivation? I would at least try a third cell type, expressing heterologous or native Piezo1 to see if the effect is generally applicable in other cell types. See also points 3 and 4.

We thank the reviewer for this comment. Because of it, we now have more data to support our claim:

1) In our original manuscript, the ratio of peak to steady-state currents of Piezo1 in HMVEC supplemented with 100 μ M EPA was not statistically significant from the control. However, the mean for cells EPA-treated was smaller than the control (41% vs. 34%, respectively). To further enhance the effect of EPA on Piezo1, we supplemented HMVEC with 50 μ M EPA for three consecutive days and found that Piezo1 inactivation was more pronounced and significantly different when compared to control HMVEC. We now have included this new data on Fig. 5e-f and Supplementary Figure 6. This result supports that EPA enhances Piezo1 inactivation in HMVEC. Furthermore, we now show with LC-MS analysis that EPA indeed accumulates at the plasma membrane after the new serial supplementation protocol. This cumulative effect is not exclusive to EPA in HMVEC, as we also supplemented N2A cells with MA and EPA for several days at low concentrations and observed a significant effect as well (Fig. 1c-d and Supplementary Figures 2a-c).

2) To address the reviewer's concern, we have followed this advice and transfected human Piezo1 in a third cell type (HEK-293) and measure Piezo1 wild type and mutants in control and EPA-treated conditions. Supplementary Figure 8b-e shows that the time constants of inactivation of Piezo1 channels in HEK-293 treated with EPA are significantly faster than those of the control. We hope these

new results ameliorate this reviewer's concern about our claim regarding the effect of EPA in Piezo1 in different cell types.

3- Free fatty acids in the blood plasma bind to proteins, mainly albumin. Protein concentration in the plasma, where red blood cells are located, is ~70g/L, which is much higher than in the interstitium and in standard cell culture media. I am not sure if the same concentrations of fatty acids would have the same effect if they were dissolved in high protein buffer; this is easy to test, and I recommend that the authors address this.

We understand the concern of the reviewer and we would like to provide three lines of evidence to diminish it:

1) Our culture media approximately contains 35 g/L of protein (according to Thermo Fisher's certificate of analysis), an amount that is below the plasma protein concentrations (as pointed out by the reviewer). Therefore, we added 70 g/L of BSA (the most abundant protein in human blood plasma) to our media while supplementing with EPA. As shown on Supplementary Figure 8f-g, high protein concentration (~105 g/L) does not impede the effect that EPA has on Piezo1 inactivation as its currents become significantly faster.

2) Daak and collaborators demonstrated that treatment with ω -3 fatty acids in humans significantly increased blood cell membrane concentration of DHA and EPA after treatment (Daak et al., 2018; Daak et al., 2013). This was achieved by providing doses as low as 20 mg/kg per day of ω -3 fatty acids to children between 5 to 17 years old and significantly increasing ($p < 0.01$) the content of EPA and DHA in red blood cell membranes. Considering that 5-year-olds weight on average 18 kg and approximately have 1.4 L of blood, we did a back-of-the-envelope calculation and found that these subjects were receiving about 300 μ M of fatty acids per day. This suggests that our supplementation concentrations are within the range of what would be effective, even in the presence of albumin.

3) To "mimic" the aforementioned clinical trial, we included in the manuscript another supplementation protocol where we provided lower concentrations of fatty acids for several days (e.g., 10-50 μ M for 3-5 days) and found that the effect in current densities and time constant of inactivation for MA and EPA, respectively, were different from the control (see Figs. 1c-d, Fig. 5e-f, and Supplementary Figure 2a-c, and Supplementary Figure 6). LC-MS analyses demonstrated that, indeed, these fatty acids accumulated in the membrane using lower supplementing concentrations for several days.

Having said this, we do not pretend that our simplistic cell culture supplementation will serve as a measure of how much would be ideal for supplementing human diet but, rather, we offer the opportunity for other laboratories to consider these fatty acids for further studies.

4- The authors only tested one Piezo 1 mutant; I think the data would be stronger if they also examined other Xerocytosis mutants, to see if the effect is generally applicable.

We agree with the reviewer and thank her/him for the suggestion. We have now included four other xerocytosis mutants in the revised version of the manuscript (see Fig. 7). Of note, EPA speeds up the time constant of inactivation of the xerocytosis mutants tested (R1942Q, M2224R, R2301H, R2456H, and R2487Q), in N2A and HEK-293 cells (see Fig. 7 and Supplementary Figures 8a-e). Overall, our results suggest that enriching the plasma membrane with EPA overrides the effect of xerocytosis gain-of-function mutants that slow down inactivation in Piezo1.

5- The authors show in Figure 1 that acute perfusion of MA has no effect on Piezo1 currents. Similar data should be shown for the unsaturated fatty acids also.

We agree with the reviewer and thank her/him for the suggestion. Now we show on Fig. 1g, Fig. 2f, Supplementary Figure 1h, and Supplementary Figure 2d that perfusion of MA, AA, EPA, and DHA do not affect Piezo1 mechano-dependent currents. Importantly, we used Gd^{3+} as a positive control of our perfusion to corroborate the lack of effect of these fatty acids.

6- The authors incubate cells with various fatty acids. From the manuscript it is not entirely clear if these incorporate in the plasma membrane as free fatty acids, or they are incorporated in phospholipids, i.e. covalently bind to a glycerol backbone. This should be clarified and discussed.

We agree with the reviewer and apologize for the missing information. We support the idea that the effect of fatty acids on Piezo1 is mainly due to esterified fatty acids instead of free because when free fatty acids are perfused in patch clamp recordings, we do not observe an effect on Piezo1 activity (Fig. 1g, Fig. 2f, Supplementary Figure 1h, and Supplementary Figure 2d). In the cases of BK and pentameric ligand gated channels perfusion of free fatty acids exert a very robust effect on their function [for more references, please see (Cordero-Morales and Vasquez, 2018)]. We have now added the new results pertaining to free fatty acids perfusion (MA, AA, EPA, and DHA) to directly address this point.

7- Introduction: “Piezo1 seems to be essential in mammals” – this is correct in mice, but the loss of function mutations in humans are compatible with life.

We thank the reviewer for noticing this error. We have changed to “Piezo1 is important in mammals” on line 52, Page 3.

8- I would put supplemental Figure 2B-D to the main figures, if the authors wish to claim these fatty acids could be therapeutically useful, it is better to show that data that they do not decrease current amplitudes.

We thank the reviewer for this suggestion. Former Supplementary Figure 2B is now part of the main figures (Fig. 2c). We have removed Former Supplementary Fig. 2D, as our mechanical stimulator was not fast enough to resolve Piezo1 activation kinetics (this was rightfully pointed out by Reviewer 3 below).

Reviewer 3:

1- Tether-pulling experiments are reported with MA but not with PUFAs. Inclusion of this data is needed to fully support the claim that MA and PUFAs have opposite effects on the membrane mechanics which influences Piezo1 gating. Minor: panels on tether-pulling experiments in Fig. 2F are not labeled as to which is control and which MA-treated.

We thank the reviewer for this suggestion and now have included in the revised manuscript the mean tether forces corresponding to N2A cell membranes enriched with MA, PDA, AA, EPA, and DHA (see Fig. 3c). AA, EPA, and DHA decreased the mean tether forces of N2A plasma membranes treated with these PUFAs, as was previously demonstrated for AA and EPA in other cell types (Caires et al., 2017; Vasquez et al., 2014). Now our claim is fully supported, as MA has the opposite effect on plasma membranes than long PUFAs.

2- Does the change in mean tether force arise from changes in mechanics of the lipid membrane or from changes in the connection between the membrane and the cytoskeleton? Tether-pulling assays on nascent membrane blebs or after treatment with cytoskeletal disrupting agents would be insightful in determining how the modifying membrane composition affects Piezo1 gating.

We thank the reviewer for this suggestion. We have added new AFM and electrophysiology recordings addressing the contribution of the cytoskeleton to Piezo1 modulation by fatty acids. To determine the contribution of the cytoskeleton to the mechanics of the plasma membrane, we disrupted actin polymerization, following previously established protocols (Krieg et al., 2014). To that end, we incubated cells with latrunculin A (1 μ M) for one hour and found that this treatment did not significantly change the mean tether forces of N2A plasma membranes (61.9 \pm 23.4 control vs. 59.7 \pm 23.1 latrunculin A-treated N2A cells, mean \pm SD). Moreover, disrupting the actin filaments did not change the effect that fatty acids enrichment had on the plasma membrane, as MA increased the mean tether force (73.6 \pm 21, mean \pm SD) and EPA decreased it (41.3 \pm 15.4, mean \pm SD; Fig. 3d). As expected, MA inhibited Piezo1 currents and increased the displacement threshold, and EPA enhanced its inactivation in latrunculin A-treated cells (Supplementary Figure 3c-f).

Disrupting actin polymerization decreased the magnitude of Piezo1 current densities (-3.11 ± 1.1 control vs. -1.49 ± 0.5 latrunculin A-treated N2A cells, mean \pm SD; Fig. 1c and Supplementary Figure 3d, gray bars). This result suggests that fewer channels are being recruited with the mechanical probe in cells treated with latrunculin A, likely due to the disrupted cytoskeleton and an inefficient transmission of the mechanical stimuli. Similar results were reported by the Goodman laboratory when measuring mechanoreceptor currents and mean tether forces from neurons lacking cytoskeletal proteins (Krieg et al., 2014; O'Hagan et al., 2005). These new data are now included in our revised manuscript on Fig. 3d and Supplementary Figure 3b-f.

3- Activation time course data throughout the paper: Since the channel activates at sub-millisecond timescales which are not resolved by the standard poking assay, presenting activation tau data in different conditions is not informative. Hence, these data should be removed. If the authors have a stimulus probe that stimulates the cell fast enough to resolve Piezo1 activation kinetics, further evidence should be presented demonstrating this.

We agree with the reviewer and have removed the activation data from the manuscript.

4- Fig. S3: GdCl3 concentration reported (30 mM) is very high. Typically a 10-fold lower (30 Micromolar) GdCl3 is used. The authors should clarify the concentration used, and perform experiments at micromolar concentrations if not done.

We thank the reviewer for this suggestion. Accordingly, we repeated the experiment with 30 μ M in HMVEC and show that this concentration is enough to inhibit Piezo1 currents in HMVEC (see Supplementary Figure 4).

5- References cited: Introduction, para 1, line 4: Also add new paper on Piezo1 in blood pressure regulation by Zeng et al. Science 2018.

Thank you for the suggestion. We have updated the bibliography (line 50, page 3).

6- Introduction, para 2, line 1 (and Results, para 1, line 4): "While Piezo1 it is widely agreed upon that Piezo1 is activated by membrane tension...", also include Syeda et al. Cell Reports 2016.

Thank you for the suggestion. We have updated the bibliography (line 63, page 3).

7- Section "Piezo1 mediated indentation-activated currents in human microvascular endothelial cells", line 2: "...where it has been shown to gate in response to shear stress" also include Ranade et al PNAS 2014

Thank you for the suggestion. We have updated the bibliography (line 214, page 8).

8- Discussion paragraph 2: units of area change should be in nm²

Thank you for noticing this typo. We have corrected it on line 359 page 12.

9- Legend of Fig. 6F, line 1: Legend includes Piezo1 WT but there is no WT data in the panel

We have completely rearranged this figure (now Fig. 7 and Fig. 8) per suggestions from Reviewers 1 and 2 suggestions. As well, we made sure that the legends accurately represent the figures.

10- Results section "Fatty acids alter Piezo1 gating by changing the physical properties of the plasma membrane", sentence: "Our results suggest that although saturated fatty acids (like MA) control the extent of Piezo1 activation, once Piezo1 is open, long PUFAs determine the degree of its inactivation.": Control and treated cells in Fig. 2A show a similar extent of inactivation, but difference in inactivation time courses, thus, the "degree of inactivation" should be revised to "time course of inactivation"

We agree with the reviewer and thank her/him for noticing this error. We have corrected this on line 160, page 6.

11- It would helpful to have page numbers and line numbers in the manuscript.

Thank you for the suggestion. Page and line numbers now are included in the revised manuscript version.

References:

- Allison, T.M., Reading, E., Liko, I., Baldwin, A.J., Laganowsky, A., and Robinson, C.V. (2015). Quantifying the stabilizing effects of protein-ligand interactions in the gas phase. *Nat Commun* 6, 8551.
- Anderson, E.O., Schneider, E.R., Matson, J.D., Gracheva, E.O., and Bagriantsev, S.N. (2018). TMEM150C/Tentonin3 Is a Regulator of Mechano-gated Ion Channels. *Cell Rep* 23, 701-708.
- Bae, C., Gnanasambandam, R., Nicolai, C., Sachs, F., and Gottlieb, P.A. (2013). Xerocytosis is caused by mutations that alter the kinetics of the mechanosensitive channel PIEZO1. *Proc Natl Acad Sci U S A* 110, E1162-1168.
- Cahalan, S.M., Lukacs, V., Ranade, S.S., Chien, S., Bandell, M., and Patapoutian, A. (2015). Piezo1 links mechanical forces to red blood cell volume. *Elife* 4.
- Caires, R., Sierra-Valdez, F.J., Millet, J.R.M., Herwig, J.D., Roan, E., Vasquez, V., and Cordero-Morales, J.F. (2017). Omega-3 Fatty Acids Modulate TRPV4 Function through Plasma Membrane Remodeling. *Cell Rep* 21, 246-258.
- Cordero-Morales, J.F., and Vasquez, V. (2018). How lipids contribute to ion channel function, a fat perspective on direct and indirect interactions. *Curr Opin Struct Biol* 51, 92-98.
- Coste, B., Mathur, J., Schmidt, M., Earley, T.J., Ranade, S., Petrus, M.J., Dubin, A.E., and Patapoutian, A. (2010). Piezo1 and Piezo2 are essential components of distinct mechanically activated cation channels. *Science* 330, 55-60.
- Daak, A.A., Dampier, C.D., Fuh, B., Kanter, J., Alvarez, O.A., Black, L.V., McNaull, M.A., Callaghan, M.U., George, A., Neumayr, L., *et al.* (2018). Double-blind, randomized, multicenter phase 2 study of SC411 in children with sickle cell disease (SCOT trial). *Blood Adv* 2, 1969-1979.
- Daak, A.A., Ghebremeskel, K., Hassan, Z., Attallah, B., Azan, H.H., Elbashir, M.I., and Crawford, M. (2013). Effect of omega-3 (n-3) fatty acid supplementation in patients with sickle cell anemia: randomized, double-blind, placebo-controlled trial. *Am J Clin Nutr* 97, 37-44.
- Diz-Munoz, A., Krieg, M., Bergert, M., Ibarlucea-Benitez, I., Muller, D.J., Paluch, E., and Heisenberg, C.P. (2010). Control of directed cell migration in vivo by membrane-to-cortex attachment. *PLoS Biol* 8, e1000544.
- Ellefsen, K., Chang, A., Nourse, J.L., Holt, J.R., Arulmoli, J., Mekhdjian, A., Abuwarda, H., Tombola, F., Flanagan, L.A., Dunn, A.R., *et al.* (2018). Myosin-II mediated traction forces evoke localized Piezo1 Ca²⁺ flickers. *bioRxiv*, 28.
- Glogowska, E., Schneider, E.R., Maksimova, Y., Schulz, V.P., Lezon-Geyda, K., Wu, J., Radhakrishnan, K., Keel, S.B., Mahoney, D., Freidmann, A.M., *et al.* (2017). Novel mechanisms of PIEZO1 dysfunction in hereditary xerocytosis. *Blood* 130, 1845-1856.
- Hopper, J.T., Yu, Y.T., Li, D., Raymond, A., Bostock, M., Liko, I., Mikhailov, V., Laganowsky, A., Benesch, J.L., Caffrey, M., *et al.* (2013). Detergent-free mass spectrometry of membrane protein complexes. *Nat Methods* 10, 1206-1208.
- Krieg, M., Dunn, A.R., and Goodman, M.B. (2014). Mechanical control of the sense of touch by beta-spectrin. *Nat Cell Biol* 16, 224-233.
- Krieg, M., Helenius, J., Heisenberg, C.P., and Muller, D.J. (2008). A bond for a lifetime: employing membrane nanotubes from living cells to determine receptor-ligand kinetics. *Angew Chem Int Ed Engl* 47, 9775-9777.
- Lacroix, J.J., Botello-Smith, W.M., and Luo, Y. (2018). Probing the gating mechanism of the mechanosensitive channel Piezo1 with the small molecule Yoda1. *Nat Commun* 9, 2029.

Laganowsky, A., Reading, E., Allison, T.M., Ulmschneider, M.B., Degiacomi, M.T., Baldwin, A.J., and Robinson, C.V. (2014). Membrane proteins bind lipids selectively to modulate their structure and function. *Nature* *510*, 172-175.

Landreh, M., Marty, M.T., Gault, J., and Robinson, C.V. (2016). A sliding selectivity scale for lipid binding to membrane proteins. *Curr Opin Struct Biol* *39*, 54-60.

Lulevich, V., Zimmer, C.C., Hong, H.S., Jin, L.W., and Liu, G.Y. (2010). Single-cell mechanics provides a sensitive and quantitative means for probing amyloid-beta peptide and neuronal cell interactions. *Proc Natl Acad Sci U S A* *107*, 13872-13877.

Ma, S., Cahalan, S., LaMonte, G., Grubaugh, N.D., Zeng, W., Murthy, S.E., Paytas, E., Gamini, R., Lukacs, V., Whitwam, T., *et al.* (2018). Common PIEZO1 Allele in African Populations Causes RBC Dehydration and Attenuates Plasmodium Infection. *Cell* *173*, 443-455 e412.

Muller, D.J., Krieg, M., Alsteens, D., and Dufrene, Y.F. (2009). New frontiers in atomic force microscopy: analyzing interactions from single-molecules to cells. *Curr Opin Biotechnol* *20*, 4-13.

O'Hagan, R., Chalfie, M., and Goodman, M.B. (2005). The MEC-4 DEG/ENaC channel of *Caenorhabditis elegans* touch receptor neurons transduces mechanical signals. *Nat Neurosci* *8*, 43-50.

Pathak, M.M., Nourse, J.L., Tran, T., Hwe, J., Arulmoli, J., Le, D.T., Bernardis, E., Flanagan, L.A., and Tombola, F. (2014). Stretch-activated ion channel Piezo1 directs lineage choice in human neural stem cells. *Proc Natl Acad Sci U S A* *111*, 16148-16153.

Qi, Y., Andolfi, L., Frattini, F., Mayer, F., Lazzarino, M., and Hu, J. (2015). Membrane stiffening by STOML3 facilitates mechanosensation in sensory neurons. *Nat Commun* *6*, 8512.

Servin-Vences, M.R., Moroni, M., Lewin, G.R., and Poole, K. (2017). Direct measurement of TRPV4 and PIEZO1 activity reveals multiple mechanotransduction pathways in chondrocytes. *Elife* *6*.

Syeda, R., Xu, J., Dubin, A.E., Coste, B., Mathur, J., Huynh, T., Matzen, J., Lao, J., Tully, D.C., Engels, I.H., *et al.* (2015). Chemical activation of the mechanotransduction channel Piezo1. *Elife* *4*.

Vasquez, V., Krieg, M., Lockhead, D., and Goodman, M.B. (2014). Phospholipids that contain polyunsaturated fatty acids enhance neuronal cell mechanics and touch sensation. *Cell Rep* *6*, 70-80.

Wang, S., Chennupati, R., Kaur, H., Iring, A., Wettschureck, N., and Offermanns, S. (2016). Endothelial cation channel PIEZO1 controls blood pressure by mediating flow-induced ATP release. *J Clin Invest* *126*, 4527-4536.

Wang, Y., Chi, S., Guo, H., Li, G., Wang, L., Zhao, Q., Rao, Y., Zu, L., He, W., and Xiao, B. (2018). A lever-like transduction pathway for long-distance chemical- and mechano-gating of the mechanosensitive Piezo1 channel. *Nat Commun* *9*, 1300.

Zhao, Q., Zhou, H., Chi, S., Wang, Y., Wang, J., Geng, J., Wu, K., Liu, W., Zhang, T., Dong, M.Q., *et al.* (2018). Structure and mechanogating mechanism of the Piezo1 channel. *Nature* *554*, 487-492.

Reviewers' Comments:

Reviewer #1:

Remarks to the Author:

The revised manuscript is in excellent shape. The authors have made significant improvements in describing and displaying their data and have gone beyond my expectations with their revisions. I congratulate the authors on their work and have nothing more to add to the revised version of the manuscript. I endorse the publication as is.

Reviewer #2:

Remarks to the Author:

The authors provided a thorough and reasonable response to the critiques, therefore I recommend acceptance.

Reviewer #3:

Remarks to the Author:

The authors have adequately addressed my questions. The new section added to the beginning of the Discussion section is confusing and hard to understand. Otherwise, the manuscript has improved considerably in response to feedback from all three reviewers.